# Intratumoral synthesis of nano-metalchelate for tumor catalytic therapy by ligand field-enhanced coordination

Bowen Yang [1,2], Heliang Yao[1], Han Tian[1,2], Zhiguo Yu[1,2], Yuedong Guo[1,2], Yuemei Wang[1,2], Jiacai Yang[1,2], Chang Chen[1,2] & Jianlin Shi [1✉]

The iron gall ink-triggered chemical corrosion of hand-written documents is a big threat to Western cultural heritages, which was demonstrated to result from the iron gall (GA-Fe) chelate-promoted reactive oxygen species generation. Such a phenomenon has inspired us to apply the pro-oxidative mechanism of GA-Fe to anticancer therapy. In this work, we construct a composite cancer nanomedicine by loading gallate into a Fe-engineered mesoporous silica nanocarrier, which can degrade in acidic tumor to release the doped $Fe^{3+}$ and the loaded gallate, forming GA-Fe nanocomplex in situ. The nanocomplex with a highly reductive ligand field can promote oxygen reduction reactions generating hydrogen peroxide. Moreover, the resultant two-electron oxidation form of GA-Fe is an excellent Fenton-like agent that can catalyze hydrogen peroxide decomposition into hydroxyl radical, finally triggering severe oxidative damage to tumors. Such a therapeutic approach by intratumoral synthesis of GA-Fe nano-metalchelate may be instructive to future anticancer researches.

[1] State Key Laboratory of High Performance Ceramics and Superfine Microstructure, Shanghai Institute of Ceramics, Chinese Academy of Sciences, Shanghai 200050, People's Republic of China. [2] Center of Materials Science and Optoelectronics Engineering, University of Chinese Academy of Sciences, Beijing 100049, People's Republic of China. ✉email: jlshi@mail.sic.ac.cn

ron gall ink is one of the most widely used inks in the history of Western civilization for preparing manuscripts, documents, and musical compositions, especially in Europe through the Middle Ages until the 20th century[1]. The chelation of ferrous ions ($Fe^{3+}$) by gallic acid (3,4,5-trihydroxybenzoic acid, GA) is the key chemical reaction for preparing iron gall ink, resulting in the formation of 3D iron-gallate (GA–Fe) metalchelate polymer with deep black color for using as writing/painting materials[2]. However, such an iron coordination compound is highly reductive that can reduce oxygen ($O_2$) to superoxide anion ($O_2^{\bullet-}$) and hydrogen peroxide ($H_2O_2$), initiating subsequent Fenton-like reactions, especially in acidic environment that generate highly oxidizing hydroxyl radicals (•OH), leading to the oxidation and degradation of cellulose in papers[3,4]. The corrosion of documents along the hand-written scripts of iron gall ink would arouse serious conservation problems for historical artefacts using this ink, such as the deterioration of complete works of Victor Hugo and 60–70% of Leonardo da Vinci's oeuvre.

The development of ligand-field theory in quantum chemistry has provided useful tools for exploring the metal–ligand cooperativity in coordination compounds as well as the related redox processes[5–7]. GA–Fe is a hexacoordinated complex with a 1:1 stoichiometry of Fe/gallate[8], as well as a slightly distorted octahedral geometry due to the Jahn-Teller effect[9]. Each iron center coordinates with the phenate and carboxylate oxygens of four gallate molecules through six Fe–O bonds[10]. The pseudo-radical electronic structure of gallate makes its five oxygen atoms electron donors favoring ligand-to-metal reduction[11]. The strong metal–ligand exchange coupling between Fe center and gallate ligands results in a strong electronic delocalization throughout the whole polymer, which not only significantly enhances light absorption to show deep black color of the iron gall ink, but also makes the spin densities of ion centers closer to the values of high-spin iron(II)[11]. Therefore, the ion centers act as active sites of the metallo-gallate complex that can donate electrons from the gallate ligands to ambient free oxygen molecules for the oxygen reduction and reactive oxygen species (ROS) generation[3], after which the ligand field around iron center attenuates due to the electron loss from gallate[12].

Interestingly, GA is also a key bioactive component of green tea and Yunnan pu-erh tea[13,14], and GA extracts from tea leaves have been demonstrated multiple biological and pharmaceutical functions, such as anti-fatigue, antibacterial, and anticancer properties[15]. The anticancer effect of GA may result from its complexation with labile iron ions in cancer cells[16], after which the chelate compound GA–Fe promotes ROS generation in acidic tumor region[17]. However, the excessive uptake of GA will result in the reduced systemic iron availability by binding with non-heme iron species[18], and heavy tea drinkers may suffer the risk of iron deficiency and anemia[19]. Therefore, to develop the anticancer application of GA or GA–Fe, the coordination reaction between GA and $Fe^{3+}$ should be regulated to take place exclusively in tumor region. To further enhance the antitumor efficiency, the intratumoral concentrations of GA and $Fe^{3+}$ should be further largely elevated after extrinsic supplementation of the two chemicals.

The coordination reaction between GA/gallate and $Fe^{3+}$ is spontaneous and kinetically fast (rate constants for $k_1 = 2.83 \times 10^3\ M^{-1}\ s^{-1}$, $k_{-1} = 20\ M^{-1}\ s^{-1}$)[20]. Therefore, it is a feasible strategy to design a drug delivery system (DDS) co-loaded with both GA/gallate and $Fe^{3+}$ but separately in different locations from each other, which then can release the two chemicals concurrently to enable the coordination reaction between them specifically in tumor region for the synthesis of iron gall complex in situ. This strategy, if applicable, can significantly elevate the oxidative damage against tumors with negligible systemic side

effect by positioning the oxygen reduction reactions (ORRs) in tumor region. Mesoporous silica nanoparticles (MSNs) have been extensively explored as DDSs due to their unique mesoporous structure, abundant surface chemistry, and good biocompatibility[21]. More importantly, their –Si–O–Si– framework of MSNs can be engineered by doping with metallic elements to form a –Si–O–M– (M = Fe, Cu, Mn, Ca, Mg) hybrid framework, which enables much improved biodegradability as well as multifunctionality[22].

In this work, we report the construction of such a nanomedicine for cancer therapy by loading gallate in an Fe-engineered hollow MSN functionalized with polyethylene glycol (PEG) (Fe-HMSN-PEG-gallate, denoted FHPG) (Fig. 1). The mesoporous and hollow structures of FHPG enable efficient gallate loading, while the unique –Si–O–Fe– hybrid framework of the nanocarrier can degrade in tumor region specifically in response to mild acidic environment of the tumor to release $Fe^{3+}$ and gallate, after which the two chemicals coordinate with each other spontaneously to form a nano-dimensional hexacoordinated GA–Fe complex in situ. The strong metal–ligand exchange coupling between Fe center and gallate ligands in GA–Fe leads to significant electronic delocalization over the whole coordination compound, making the nanoparticle a qualified electron donor. The unique ligand field around Fe centers in the pseudo-octahedral structure of GA–Fe polymer promotes the two-electron reduction of $O_2$ into $H_2O_2$, after which the metal–ligand interaction becomes weakened due to the partial oxidation of gallate ligand. Importantly, the two-electron oxidation form of GA–Fe nanoparticles with attenuated ligand field can further catalyze Fenton-like reactions efficiently to promote the generation of highly oxidizing •OH, finally triggering oxidative damage to tumor. Cellular experiments and in vivo model further demonstrate the high anticancer efficacy of FHPG by intratumoral synthesis of iron gall nano-metalchelate, indicating the feasibility of such a catalytic therapeutic approach for future cancer therapy.

## Results

**Synthesis and characterization**. The synthesis of Fe-engineered hollow MSNs (Fe-HMSNs) was based on a hydrothermal reaction approach using pristine MSNs as hard templates (Fig. 2a)[23]. Pristine MSNs were first prepared via a typical sol-gel approach by the condensation of silica precursor tetraethyl orthosilicate (TEOS) using cetyltrimethylammonium chloride (CTAC) surfactant as a structure-directing agent and triethanolamine (TEA) as an alkaline catalyst. Transmission electron microscopy (TEM) image and selected area electron diffraction (SAED) pattern indicate that the as-prepared MSNs are monodispersed with an amorphous structure, (Supplementary Fig. 1), favoring subsequent preparation of Fe-HMSNs in a harsh hydrothermal basic condition, during which the –Si–O–Si– framework of MSN template gradually hydrolyzes into Si-containing oligomers such as orthosilicic acid ($Si(OH)_4$) on the surface of MSNs. These oligomers can bond with iron ions from the added Fe precursor iron acetylacetonate ($Fe(acac)_3$), to form an iron silicate layer, and the further alkali etching of inner MSN template and the further reaction between the released Si-containing oligomers and exogenous $Fe(acac)_3$ finally results in the generation of a hollow nanostructure with a –Si–O–Fe– hybrid framework.

TEM images indicate that the as-prepared Fe-HMSNs are monodispersed with a uniform diameter of around 200 nm (Fig. 2b), which is further evidenced by dynamic light scattering (DLS) measurement (Supplementary Fig. 2). These nanoparticles show distinct hollow structure with abundant pore channels in the framework (Fig. 2c), as well as a rough surface topography

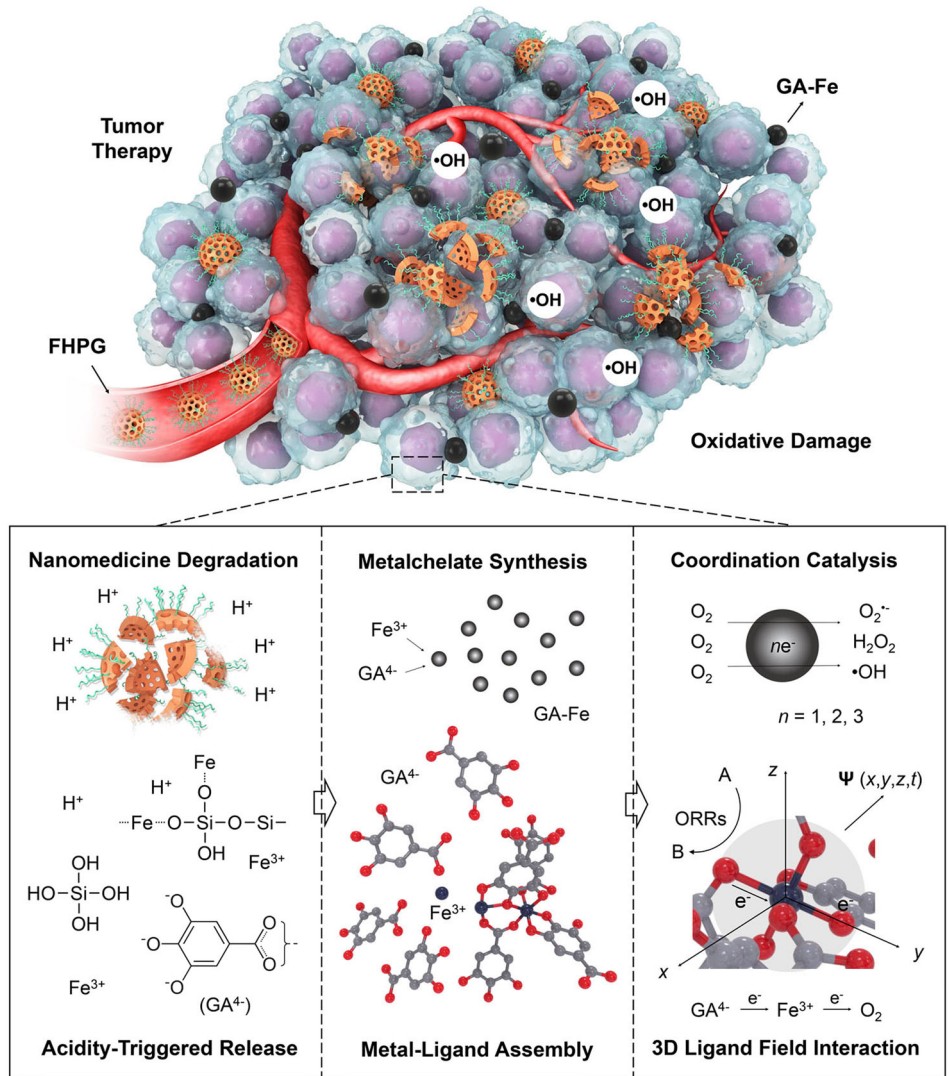

**Fig. 1 Chemical mechanism schematics for the nanomedicine FHPG for tumor therapy.** After accumulation into tumor region and internalization by cancer cells, the acidic environment enables the degradation of Fe-HMSN-PEG nanocarrier by breaking the Fe–O bond in the –Si–O–Fe– hybrid framework, promoting the co-releases of framework-doped $Fe^{3+}$ and hollow core-loaded gallate ($GA^{4-}$) in the nanocarrier into the medium. Thereby, the released $GA^{4-}$ will chelate free $Fe^{3+}$ spontaneously to form a nano-dimensional coordination complex in situ, which is a qualified electron donor due to the strong metal–ligand coordination. The unique ligand field around the Fe centers of GA–Fe nanoparticles promote ORRs and generate highly oxidizing •OH, finally triggering oxidative damage to tumor.

(Fig. 2d), while $N_2$ adsorption–desorption isotherms and pore-size distribution data further reveal the well-defined mesoporous structure (Supplementary Fig. 3). High-resolution TEM image and SAED pattern manifest the weak crystallinity of Fe-HMSNs (Fig. 2e, f), in agreement with the data of X-ray diffraction (XRD) pattern (Supplementary Fig. 4). According to the results of $^{29}Si$ solid-state magic angle spinning nuclear magnetic resonance (MAS-NMR) (Supplementary Fig. 5), the distinctive peaks at chemical shifts of −90 ppm ($Q^2$, $Si(OSi)_2(OH)_2$), −100 ppm ($Q^3$, $Si(OSi)_3(OH)$), and −110 ppm ($Q^4$, $Si(OSi)_4$) of pristine MSN sample become much weakened in the spectrum of Fe-HMSN, as a result of Fe element doping within the –Si–O–Si– framework that reduces the condensation degree of silica. Element mappings and energy-dispersive spectroscopy (EDS) profile of Fe-HMSN sample demonstrate a homogeneous Fe element distribution within the nanoparticle with a rather high Fe-doping concentration (Fig. 2g and Supplementary Fig. 6). X-ray photoelectron spectroscopy (XPS) spectrum of Fe-HMSN further indicates that most of the Fe species are trivalent (Supplementary Fig. 7), while

the existence of minor quantity of divalent species may be due to the slight overdose of the Fe precursor $Fe(acac)_3$ during the hydrothermal reaction for Fe-HMSN synthesis, which leads to incomplete condensation of few Fe species with Si-containing oligomers.

The nanomedicine FHPG was fabricated by further PEGylation on Fe-HMSNs and subsequent gallate loading (Supplementary Fig. 8). Fe-HMSNs are unstable in saline solutions such as phosphate buffer saline (PBS) and will form large aggregates among each other (Supplementary Fig. 9). Therefore, to improve the stability and dispersity of the nanocarrier in saline solution environment for subsequent biological application, methoxy PEG silane was covalently modified on the surface of Fe-HMSNs, after which a new –Si–O–Si– bond formed between the particle and PEG. Fourier transform infrared (FTIR) spectra reveal the presence of an absorption peak at 2887 $cm^{-1}$ in the spectrum of modified Fe-HMSNs (Fig. 2h), which is attributed to the stretching vibration of –$CH_2$– in the carbon skeleton of PEG, demonstrating successful PEGylation. The loading of gallate by

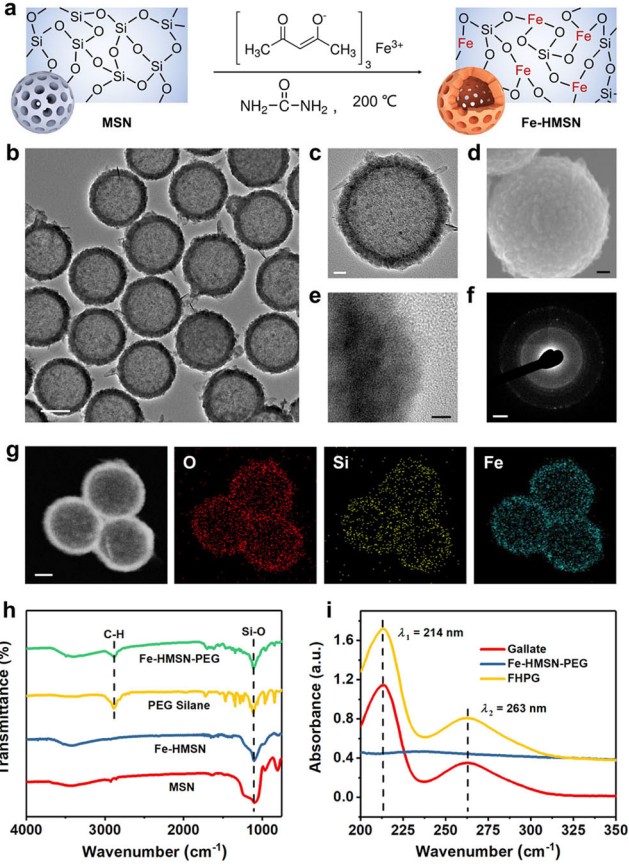

**Fig. 2 Characterizations of Fe-HMSN and FHPG. a** Chemical process of the structural and compositional evolutions of Fe-HMSN from MSN template. Fe(acac)$_3$ and urea are iron and alkaline precursors, respectively. **b** TEM image of monodispersed Fe-HMSNs. Scale bar, 100 nm. **c** TEM image of a single Fe-HMSN. Scale bar, 20 nm. **d** SEM image of a single Fe-HMSN. Scale bar, 20 nm. **e** High-resolution TEM image at the edge of a single Fe-HMSN. Scale bar, 5 nm. **f** SAED pattern of Fe-HMSN. Scale bar, 2 nm$^{-1}$. **g** High-angle annular dark-field (HAADF) image and element mappings of Fe-HMSNs. Scale bar: 50 nm. A representative image of three replicates is shown. **h** FTIR spectra of MSN, Fe-HMSN, PEG silane and Fe-HMSN-PEG. **i** UV-Vis absorption spectra of gallate, Fe-HMSN-PEG, and FHPG. Source data are provided as a Source Data file.

Fe-HMSNs-PEG was further investigated by spectrophotometric assay, as two characteristic absorption peaks of gallate can be observed in UV-Vis absorption spectra (Fig. 2i). The intense absorption at 214 nm is assigned to a $\pi$–$\pi^*$ transition in the benzene ring of gallate[24], while the absorption at 263 nm is associated with an electron transfer from the $\pi^*$-orbital of the carbonyl moiety of gallate to the aromatic $\pi$-system[25]. Although the unique mesoporous and hollow structures of Fe-HMSNs endow the nanocarrier with high drug loading efficiency, however, given the 1:1 stoichiometry of Fe/gallate in GA–Fe complex[26], in this work we also kept the stoichiometric compositions of Fe/gallate in FHPG nanomedicine to be 1:1, as excess free gallate in the absence of Fe ions would present antioxidative property[27], which might reduce the pro-oxidative capability of the nanosystem.

**Degradation and GA–Fe nano-metalchelate formation**. The Fe-doped framework and low condensation degree of silica endow the nanocarrier with abundant defects and therefore improved degradability. In acidic environment, as the energy of Fe–O bond

is lower than that of Si–O bond[28], the H$^+$ infiltration in the framework of Fe-HMSNs will promote the breakage of Fe–O bond, during which H$^+$ will substitute for Fe$^{3+}$ to promote the Fe$^{3+}$ release and the formation of new Si–OH groups (Fig. 3a). Consequently, vacancy defects could be generated in the original lattice points of Fe, which would promote the disintegration of the defective framework into Si-containing oligomers such as Si (OH)$_4$ under the assistance by the reactive Si–OH groups. We first investigated the degradability of Fe-HMSNs-PEG by dispersing them in simulated body fluid (SBF) of varied pH values. TEM images indicate that the nanocarriers have significantly degraded after immersing in a mildly acidic SBF (pH = 6.5) for 6 h, and become debris and loss their original morphology in 12 h of degradation (Fig. 3b and Supplementary Fig. 10). It was observed that these nanoparticles became completely degraded in acidic SBF in 60 h of immersion. In comparison, Fe-HMSNs-PEG still kept their morphological integrity after immersing in a neutral SBF (pH = 7.4) for 6 h and slightly degraded in 12 h of immersion (Fig. 3c and Supplementary Fig. 10). Even in 60 h of degradation, only a part of these nanoparticles has collapsed in the neutral SBF. These results demonstrate that the as-fabricated nanocarrier is highly sensitive to acidic environment which triggers its degradation in response to the acidity to release Fe$^{3+}$ as well as Si component (Fig. 3d and Supplementary Fig. 11).

On the basis of the acidity-responsive degradability of Fe-HMSNs-PEG, we then investigated the evolution of the composite nanomedicine FHPG (with gallate loading) in SBF of different pH values. In acidic environment, the co-released Fe$^{3+}$ and gallate from the nanocarrier will react with each other to form GA–Fe complex (Fig. 3e), as evidenced by the deep black color of mild acidic SBF (pH = 6.5) in 2 h of FHPG degradation (Fig. 3f), The color of the solution is analogous to that of iron gall ink capable of being used for handwriting. Spectrophotometric data indicate a bathochromic shift of the absorption band of gallate at 263 nm during 20 h of FHPG degradation in acidic environment (Fig. 3g), demonstrating the chelation of Fe$^{3+}$ by gallate[2]. According to Fig. 2i, the baseline of FHPG is attributed to the full spectrum absorption of the nanocarrier. Therefore, the baseline downshift in Fig. 3g further reveals the chelating-induced GA–Fe formation following the degradation of Fe-HMSNs-PEG nanocarrier.

According to TEM and scanning electron microscopy (SEM) images, after degradation in mild acidic SBF (pH = 6.5), only a fragment structure of FHPG could be observed, surrounded by numbers of small nanoparticles with high dispersity (Fig. 3h and Supplementary Figs. 12 and 13). The debris of FHPG shows weaker crystallinity compared with pristine Fe-HMSNs (Fig. 3i, j), as a consequence of vacancy defect generation during degradation. Interestingly, these surrounding small nanoparticles of approximately 5–20 nm in sizes (Fig. 3k) are well crystallized and distinct lattice fringe could be observed (Fig. 3l), demonstrating that these nanoparticles are not the fragments from nanocarrier degradation, but newly formed substance, i.e., GA–Fe complex. After FHPG degradation for 60 h in mild acidic SBF, only the newly formed tiny nanoparticles could be visualized (Supplementary Fig. 14), indicating the complete evolution of FHPG to GA–Fe nano-metalchelate. Comparatively, negligible degradation of FHPG nanoparticles could be observed after being dispersed in neutral SBF (pH = 7.4) for 60 h (Supplementary Fig. 15), and only a very few tiny nanoparticles could be found during the process, demonstrating the acidity specificity of FHPG degradation favoring the synthesis of iron gall coordination complex preferentially in acidic environment.

As the degradation of FHPG is a relative slow process even in acidic environment, the newly formed GA–Fe nanocomplex during the degradation will be inevitably oxidized. Therefore, to

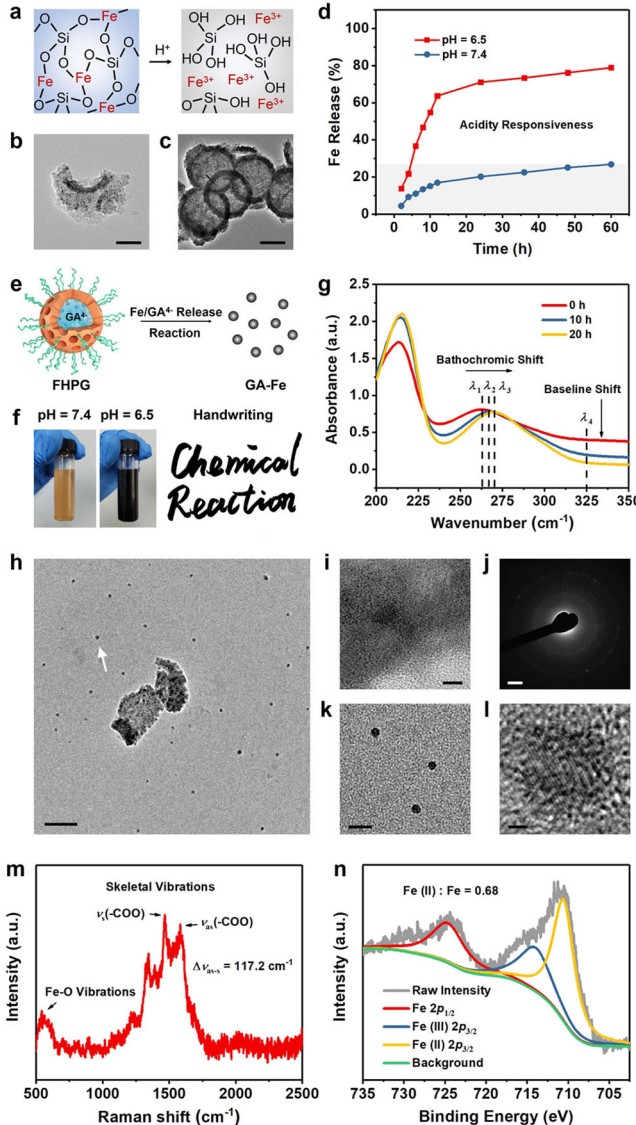

**Fig. 3 Acidity-triggered FHPG degradation and GA–Fe nano-metalchelate formation. a** Chemical mechanism for the degradation of the nanocarrier Fe-HMSN in acidic environment. The $H^+$ infiltration enables the breakage of Fe—O bond in the –Si–O–Fe– hybrid framework to promote the release of free $Fe^{3+}$ and $Si(OH)_4$. **b, c** TEM images of Fe-HMSNs-PEG after degradation in SBF for 12 h at pH = 6.5 (**b**) or 7.4 (**c**). Scale bars, 100 nm. A representative image of three replicates from each group is shown. **d** Time-dependent Fe release from Fe-HMSNs-PEG in SBF at different pHs. **e** Scheme for the evolution of FHPG to GA–Fe nanoparticles in acidic environment by $Fe^{3+}$ and GA co-releases and the subsequent chelating reaction between them. **f** Digital photos of FHPG dispersed in SBF of different pHs for 2 h, as well as the handwriting "Chemical Reaction" by using the black solution (can be considered as iron gall ink) and a Chinese writing brush. **g** UV-Vis spectra of FHPG after degradation in SBF (pH = 6.5) for different time durations. The bathochromic shift from $\lambda_1$ to $\lambda_3$ indicates the chelation of $Fe^{3+}$ by gallate and the formation of GA–Fe, while the baseline shift at $\lambda_4$ indicates the degradation of Fe-HMSNs-PEG. **h** TEM image of FHPG sample after degradation in SBF for 12 h (pH = 6.5). The white arrow indicates the newly formed GA–Fe nano-chelates of high dispersity accompanied by FHPG degradation. Scale bar, 100 nm. **i** High-resolution TEM image at the edge of Fe-HMSN-PEG piece undergoing degradation. Scale bar, 5 nm. **j** Corresponding SAED pattern of (**i**). Scale bar, 2 nm$^{-1}$. **k** TEM image of GA–Fe nanoparticles. Scale bar, 25 nm. **l** High-resolution TEM image of a single GA–Fe nanoparticle. Scale bar, 1 nm. A representative image of three replicates is shown. **m** Raman spectrum of fresh GA–Fe showing its chemical bonds and functional groups. **n** Fe 2p spectrum of XPS spectra of fresh GA–Fe. Source data are provided as a Source Data file.

fully investigate the chemical characteristics of pristine GA–Fe, fresh GA–Fe nanocomplex was prepared by the direct reaction between $Fe^{3+}$ and gallate (Supplementary Fig. 16). The metal–ligand interactions in GA–Fe were revealed by Raman spectrum of such a fresh GA–Fe complex (Fig. 3m), in which bands in the low-frequency region (650–500 cm$^{-1}$) can be assigned to Fe-O vibration as a consequence of Fe–O bonding due to the Fe chelation by gallate[29]. In addition, bands at Raman shifts of 1581.2 and 1464.0 cm$^{-1}$ can be attributed to the asymmetric ($v_{as}$) and symmetric ($v_s$) stretching of the carboxylate group (–COO) in gallate, respectively[30], while the difference between the two bands ($\Delta v_{as-s}$) is 117.2 cm$^{-1}$, manifesting a bridge-type coordination between the carboxylate oxygens of gallate ligands and Fe atoms in the GA–Fe complex (Supplementary Table 1)[31], in consistence with the configuration of a real iron gall structure[11]. To investigate the redox status of Fe center in pristine GA–Fe coordination compound, the main peak in the Fe 2p XPS spectrum of fresh GA–Fe sample was split into two sub-peaks assigned to Fe (II) $2p_{3/2}$ and Fe (III) $2p_{3/2}$ (Fig. 3n), in which the ferrous one accounts for 68% peak area of the pristine peak. Though this result cannot be simply interpreted as the conversion of 68% $Fe^{3+}$ to $Fe^{2+}$ after coordination reaction with gallate during GA–Fe formation, it is reasonable to propose

that under the effect of ligand field created by the coordinated gallate ligands, the Fe center possess 68% redox properties of divalent iron, which endows the GA–Fe nano-metalchelate with high reducibility.

**Ligand-field-enhanced chemical reactions.** Before further experimental investigation on the redox properties of GA–Fe complex, we should obtain a better understanding on the coordination chemistry of GA–Fe and the functionalities of ligand field in regulating chemical reactions, based on the important conclusions from previous literatures on iron gall complex. Wunderlich and co-workers first provided an exact demonstration on the molecular structure of GA–Fe in 1991[8]. This iron coordination compound is a 3D periodic polymer where iron ions are bridged by gallate molecules (Supplementary Fig. 17 and Supplementary Table 2). Each Fe center is hexacoordinated with a pseudo-octahedral symmetry due to Jahn-Teller distortion (Fig. 4a), in which two gallate ligands are mutually *cis*-positioned and coordinate iron with two adjacent phenate-type oxygens, while the other two ligands coordinate iron center with one carboxylate oxygen and the two oxygen atoms are also mutually in *cis* position. Each carboxylate acts as a bridging ligand connecting two iron centers (i.e., bridge-type coordination) (Fig. 4b)[11], while the three phenate oxygens on each gallate molecule chelate two iron ions. The 4-phenate oxygen in *para* position with respect to carboxylate group is a bridging atom connecting the two chelated iron ions[29].

Ligand-field theory is developed by combining crystal-field theory and molecular-orbital theory for better interpretation on the physicochemical properties of coordination compound, which is also beneficial for better understanding the electronic structure of GA–Fe in this work. It is noted that the hexacoordinated GA–Fe complex does not obey 18-electron rule (effective atomic number rule) for an ideal coordination compound. The π-system

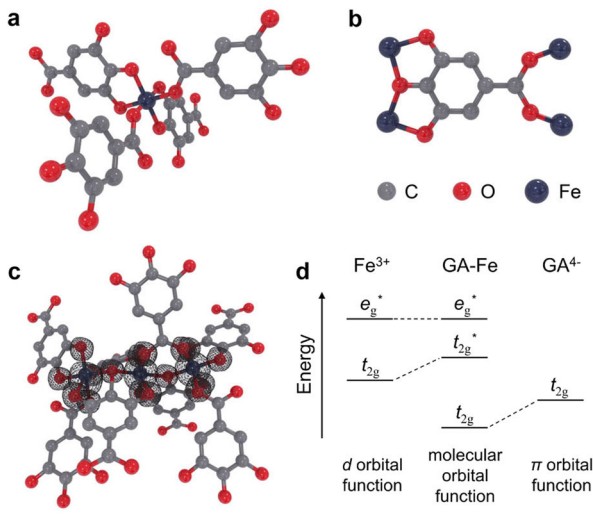

$$nFe^{3+} + nGA^{4-} \longrightarrow [Fe(III)GA]_{(x,y,z)}^- \quad \iiint dxdydz = n \quad (1)$$

$$[Fe(III)GA]_{(x,y,z)}^- \longrightarrow [Fe(II)GA\bullet]_{(x,y,z)}^- \quad (2)$$

$$[Fe(II)GA\bullet]_{(x,y,z)}^- + nO_2 \longrightarrow [Fe(III)GA\bullet]_{(x,y,z)}^- + nO_2^{\bullet-} \quad (3)$$

$$[Fe(III)GA\bullet]_{(x,y,z)} \longrightarrow [Fe(II)GA_{ox}]_{(x,y,z)} \quad (4)$$

$$[Fe(II)GA_{ox}]_{(x,y,z)} + nO_2^{\bullet-} + 2nH^+ \longrightarrow [Fe(III)GA_{ox}]_{(x,y,z)}^+ + nH_2O_2 \quad (5)$$

**Fig. 4 Coordination chemistry of GA–Fe and ligand-field-enhanced chemical reactions. a** Schematics of the octahedral coordination of four gallate molecules to an Fe center in GA–Fe. **b** Chelation of four iron centers by one gallate molecule in GA–Fe. **c** Representative tri-nuclear model of GA–Fe, $[Fe_3L_8H_{22}]^-$, as well as the distribution of spin density indicating the interaction between Fe center and gallate ligand forming ligand field. **d** Scheme for the interaction between $d$ ($t_{2g}$) orbital function of $Fe^{3+}$ with the $\pi$-orbital function of $GA^{4-}$ leading to the formation of two new molecular orbital functions of GA–Fe. **e** Proposed chemical reactions for the formation of GA–Fe nanoparticles and subsequent sequential oxygen reduction forming $O_2^{\bullet-}$ and $H_2O_2$. GA$\bullet$ indicates the semiquinone formed after one-electron oxidation of $GA^{4-}$; $GA_{ox}$ is a two-electron oxidation product of $GA^{4-}$. **f** Molecular structures of gallate and its one and two-electron oxidation products. As the electronic delocalization can occur within the whole gallate molecule, the loss of two electrons will weaken all the C–O bonds in a whole gallate molecule when coordinated with Fe and the Fe–O bonds as well, rather than forming an $o$-quinone with two carbon–oxygen double bonds which may triggers the breakage of Fe–O bond.

of the carboxylate group of gallate makes the two C–O bonds identical with each other and have a bond order of 3/2, therefore the bond order of two Fe–O bonds between the carboxylate group and two iron centers will be only 1/2. Additionally, the bond order of Fe–O bond between 4-phenate oxygen and two iron centers are also only 1/2. Therefore, in this hexacoordinated complex the electron orbitals of Fe centers are not saturated. Comparatively, the gallate ligand of $C_{2v}$ point group symmetry is rich in highly delocalized unpaired electrons, due to the unique

$p$–$\pi$ conjugation between three phenolic hydroxyl oxygen atoms and the aromatic ring, as well as the $\pi$–$\pi$ conjugation between carboxylic anion and aromatic ring. Consequently, the delocalized electrons of gallate ligand will have a considerably large possibility to be shared by ion centers. For a GA–Fe nanoparticle, a strong electronic delocalization can occur within the whole 3D periodic polymer bridged by metal centers and gallate ligands.

Based on the density functional theory (DFT) calculations by Zaccaron et al.,[11] the spin density of GA–Fe complex is localized mainly on the iron centers and to a lesser extent on the bonded phenate and carboxylate oxygens (Fig. 4c and Supplementary Table 3). The pseudo-radical electronic structure of gallate enables a ligand-to-metal reduction, making the metal center in a high-spin state largely analogous to that of $Fe^{2+}$. These Fe centers can be considered as active sites with high reducibility, while the surrounding ligand field as a potential field guides electron to flow from gallate to Fe in support of the reduction reaction on metal sites. According to molecular-orbital theory, during the coordination reaction forming GA–Fe complex, the interaction between the $d$ ($t_{2g}$) orbital function of $Fe^{3+}$ and the $\pi$-orbital function of gallate leads to the formation of two new molecular orbital functions of GA–Fe, i.e., low-energy bonding orbital function ($t_{2g}$) and high-energy antibonding orbital function ($t_{2g}^*$) (Fig. 4d). As the energy of $\pi$-orbital of gallate ligand is lower than that of $d$ orbital of $Fe^{3+}$, fortunately the unpaired electrons of gallate is abundant and delocalized, the electrons of $\pi$-orbital of the ligand will occupy the newly formed $t_{2g}$ bonding orbital, thereby the electrons of $d$ orbital of $Fe^{3+}$ can only occupy the $t_{2g}^*$ antibonding orbital, making gallate an electron donor favoring the reduction reactions on Fe centers, during which the ligand is oxidized assisted by the metal center as an electron transporter.

Abdel-Hamid et al. have proposed the oxidation mechanism of gallate (denoted $GA^{4-}$)[32], which undergoes the first one-electron oxidation to form semiquinone radical ($GA\bullet^{3-}$), while a second one-electron oxidation reaction enables the generation of $o$-quinone ($GA_{ox}^{2-}$). These oxidations of gallate are irreversible[32]. Therefore, the ligand field will undergo continuous decay during the reduction reaction on Fe centers. Based on the previous work on the oxygen-reduction capability of iron–polyphenol complexes[33], here we propose a mechanism of two sequential one-electron ORRs triggered by GA–Fe complex (Fig. 4e). The gallate ligand first enables one-electron reduction of an Fe (III) center to Fe (II), which further transfers the electron to $O_2$ to produce $O_2^{\bullet-}$. After the process Fe (III) is regenerated while the one-electron oxidation product of gallate (i.e., the semiquinone radical $GA\bullet^{3-}$) will further donate one electron to Fe (III), generating the $o$-quinone $GA_{ox}^{2-}$ and Fe (II). This Fe(II) species then transfers the electron to $O_2^{\bullet-}$ forming $H_2O_2$, accompanied by the Fe(III) regeneration again. Such an iron cycling favors the gallate–Fe–oxygen electron flux, which is associated with the intermolecular electron transfer in gallate ligand that enables continued electron supply to metal site[34].

In theory the formation of $o$-quinone will lead to the release of $Fe^{3+}$ from the 3D polymer as the generation of two C–O double bonds disenables the interaction between phenate oxygens and Fe centers. However, according to Taylor's conclusion[24] and Carter's semi-empirical molecular orbital computation[35], the carboxylic anion of gallic acid tetraanion can also donate an electron to the positive $\pi$-system of aromatic ring by $\pi$–$\pi^*$ transition after two consecutive one-electron oxidations. Therefore, in the molecule benzene ring acts as an electron transporter while all the C–O single bonds will be weakened after the electron loss (Fig. 4f), which further leads to the weakening of all Fe–O bonds, instead of the breakage of two of them as $o$-quinone does[12]. The highly delocalized nature of unpaired electrons of gallate favors an

electron loss to influence the chemical bonding of whole molecule system as well as the coordination of gallate with four metal centers. Fe ions are still weakly coordinated but not freed, by surrounding six phenate and carboxylate oxygens of ligands after the two-electron loss, but the Fe–O bonds as well as the ligand field are significantly weakened. In this situation, the redox capability of the oxidized form of GA–Fe (denoted $GA_{ox}$–Fe) mainly depends on the Fe.

**Redox property investigation for GA–Fe**. Electrochemistry is a feasible method to investigate the redox capability of molecules and nanomaterials. According to the results of cyclic voltammetry (CV), no distinct reduction peak could be visualized during CV measurement of gallate solution (Supplementary Fig. 18a), indicating that the oxidation of gallate is irreversible. Comparatively, a distinct redox cycling can be observed from the CV curve of $Fe^{3+}$ solution (Supplementary Fig. 18b). We then started the successive CV measurement of fresh GA–Fe in electrolyte solution to investigate the redox behavior of this coordination complex (Fig. 5a). The CV was originally swept from +0.1 to −0.5 V, during the process no distinct reduction peak could be visualized, demonstrating that the interaction between ligand field and Fe centers endows these metal sites with enhanced reducibility that cannot further accept electrons from external environment. Five complete cycles of CV were swept subsequently, indicating a quasi-reversible character for the redox response of GA–Fe complex, i.e., the oxidation and reduction peaks are identifiable with a distinct and continuous decrease in the current. The oxidation of GA–Fe complex at the oxidation peak enables the continuous attenuation of ligand field, as well as the weakened reducibility of Fe centers, thus the metal sites can be reduced during the negative scanning of CV and consequently the reduction peak appears (Supplementary Table 4). Therefore, such a quasi-reversible redox response of GA–Fe complex is enabled by the combination of irreversible gallate ligand oxidation and the reversible Fe center redox cycling. It should be noted that negligible changes can be observed for the light absorption property of GA–Fe sample (Fig. 5b and Supplementary Fig. 19), indicating that minor degradation of GA–Fe occurred after five complete CV cycles.

Chronoamperometry (CA) is another electrochemical characterization method usually used for the determination of diffusion coefficient, here we used this method for further evaluating the redox capability of GA–Fe as it can provide constant voltage for a certain time period for completing the redox reactions (different from CV in which the voltage is always changing), and could enable the transition between two step voltages for favoring redox coupling, by observing the responses of limited current. Based on the anodic peak potential ($E_{pa}$) and cathodic peak potential ($E_{pc}$) of GA–Fe measured in CV, we first conducted the double potential step chronoamperometry (DPSCA) measurement of electrolyte solution containing fresh GA–Fe in response to seven consecutive periods of double potential step transitions (high potential: 0.90 V; low potential: 0.15 V, step frequency: 0.0187 s$^{-1}$; period: 107 s) (Fig. 5c and Supplementary Fig. 20a). Under the potential setting the GA–Fe could only be oxidized while the Fe center in the complex could not be reduced (Supplementary Table 5). Therefore, the accomplishment of this CA test will lead to the thorough oxidation of the ligand featuring a minimized ligand-field strength. The response of the limited current in the seventh cycle of CA is much faster than that in the first one (Fig. 5d), as a consequence of distinct ligand-field attenuation during successive double potential step transitions. Additionally, the limited current response in the seventh period of CA measurement of GA–Fe is

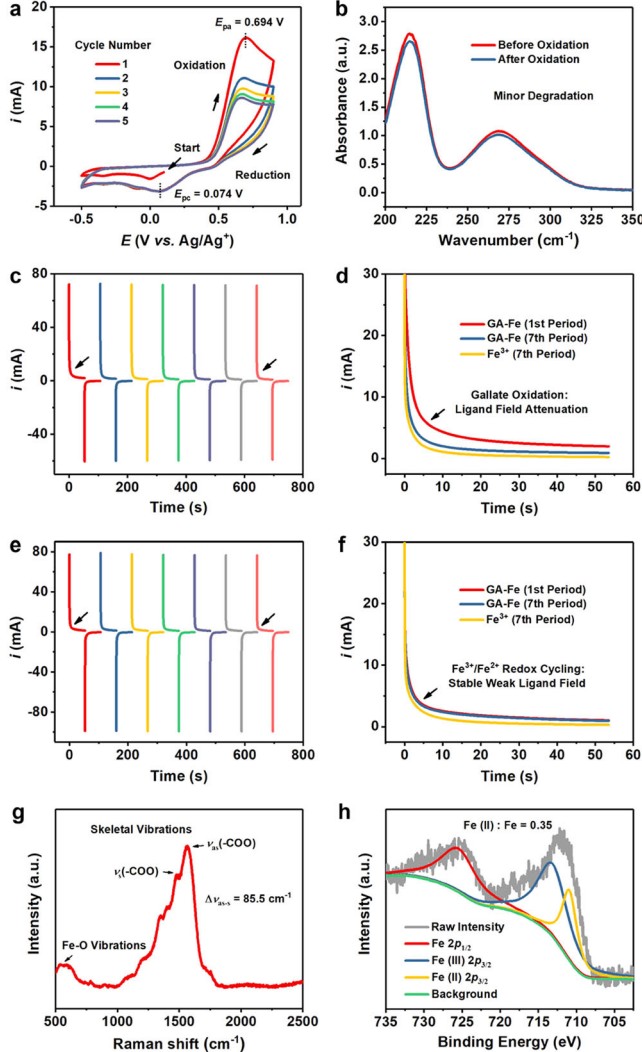

**Fig. 5 Electrochemical responses of GA–Fe and the composition of its product after electrochemical oxidation. a** Successive CV curves investigating redox behavior of fresh GA–Fe in electrolyte solution (pH = 6.5). The CV was originally swept from + 0.1 to - 0.5 V, followed by five complete cycles. **b** UV-Vis absorption spectra of GA–Fe sample before and after electrochemical oxidation enabled by successive CV scanning. **c** CA profiles of electrolyte solution containing fresh GA–Fe in response to seven consecutive periods of double potential step transitions (high potential: 0.90 V; low potential: 0.15 V). **d** Comparison among limited current responses of electrolyte solutions containing fresh GA–Fe or $Fe^{3+}$ when a half-period of continuous high potential (0.90 V) actions during the successive double potential step transitions of CA (GA–Fe: in the first and seventh periods marked in panel (**c**); $Fe^{3+}$: in the seventh period marked in Supplementary Fig. 20b). **e** The electrochemical oxidation product of GA–Fe after CA measurement in (**c**) (i.e., $GA_{ox}$–Fe) further underwent a second round of CA measurement (high potential: 0.90 V; low potential: −0.45 V; seven successive cycles). **f** Comparison among limited current responses of electrolyte solutions containing $GA_{ox}$–Fe or $Fe^{3+}$ under the action of a half-period of continuous high potential (0.90 V) during the new round of double potential step transitions (GA–Fe: in the first and seventh period marked in panel (**e**); $Fe^{3+}$: in the seventh period marked in Supplementary Fig. 20d). As this round of CA is catalytic, the limited current response of $GA_{ox}$–Fe is nearly not changed during seven consecutive periods of double potential step transitions. **g** Raman spectrum of electrochemically oxidized GA–Fe nanoparticles after two rounds of CA measurements. **h** Fe 2p spectrum of XPS spectra of electrochemically oxidized GA–Fe nanoparticles after two rounds of CA measurements. Source data are provided as a Source Data file.

close to that of $Fe^{3+}$ but the difference between them still exists (Fig. 5d and Supplementary Fig. 20b), demonstrating that the oxidation of gallate ligand and the attenuation of ligand field have weakened the interaction with Fe (III) center, making GA–Fe complex mainly present the redox characteristics of metal sites, though a weak metal–ligand interaction still exists. This process corresponds to the proposed chemical reactions in Fig. 4e, where the GA–Fe acts as an electron donor and is oxidized.

The electrochemically oxidized form of GA–Fe (i.e., $GA_{ox}$–Fe) was further used for second round of DPSCA measurement with the actions of newly established double potential step transitions (high potential: 0.90 V; low potential: −0.45 V; step frequency: $0.0187\ s^{-1}$; period: 107 s) (Fig. 5e and Supplementary Fig. 20c). Under such a potential setting the Fe center in $GA_{ox}$–Fe could also be reduced by the periodic action of low potential (−0.45 V) (Supplementary Table 6). Therefore, a continuous $Fe^{3+}/Fe^{2+}$ redox cycling will occur in the coordination compound, making the redox cycling catalytic to oxidation of the surrounding substances. The response characteristic of limited current is almost negligibly changed during the seven successive double potential step transitions (Fig. 5f), further evidencing the catalytic property of $GA_{ox}$–Fe in this new round of CA. Additionally, the limited current responses during the CA measurement of $GA_{ox}$–Fe is close to, but not fully identical with that of $Fe^{3+}$ (Fig. 5f and Supplementary Fig. 20d), indicating that the $GA_{ox}$–Fe presents the redox behavior analogous to that of $Fe^{3+}$, though a relative weak ligand–field interaction still exist.

Notably, during the two rounds of CA measurements, insignificant $Fe^{3+}$ release from the nanoparticles was detected (Supplementary Table 7), demonstrating the structural integrity of the nanocomplex after electrochemical oxidation. The $GA_{ox}$–Fe after the second round of CA measurement was collected and characterized. The metal–ligand interactions in $GA_{ox}$–Fe were revealed by Raman spectrum (Fig. 5g), in which the difference between the bands for the asymmetric and symmetric stretching of carboxylate ($\Delta v_{as-s}$) is only $85.5\ cm^{-1}$, demonstrating a bidentate coordination mode[30]. According to the Fe 2p XPS spectrum of $GA_{ox}$–Fe (Fig. 5h), the split Fe (II) $2p_{3/2}$ peak accounts for 35% area of the main peak, indicating that the metal center only possess 35% redox property of divalent iron after the attenuation of ligand field. TEM image of $GA_{ox}$–Fe shows its similar size and morphology to that of GA–Fe (Supplementary Fig. 21a), while high-resolution TEM image of $GA_{ox}$–Fe further indicates its well-crystallized nature (Fig. 6a). However, XRD patterns show a distinct crystallographic transformation of GA–Fe after electrochemical oxidation (Supplementary Fig. 21b), suggesting a new phase formation. Collectively, we propose the chemical structure of the coordinated iron center in $GA_{ox}$–Fe (Fig. 6b), which is weakly bonded with six oxygen atoms from three oxidized gallate ligands, two of the ligands chelate the metal center, respectively, with their two phenate-type oxygens, while the other ligand coordinates with the Fe center by two oxygen atoms of carboxylate group in a bidentate mode.

Based on the results of second round of CA, the redox property of $GA_{ox}$–Fe has been identified to be analogous to that of $Fe^{3+}$, therefore, we then investigated the catalytic capability of the $GA_{ox}$–Fe nanoparticles toward $H_2O_2$ decomposition to •OH (Fenton-like reactions). Electron spin resonance (ESR) spectrum of mild acidic buffer solution (pH = 6.5) containing $GA_{ox}$–Fe shows a characteristic 1:2:2:1 •OH signal after $H_2O_2$ addition (Supplementary Fig. 22), while no •OH signal could be observed in the spectrum of neutral buffer solution (pH = 7.4) containing $GA_{ox}$–Fe and $H_2O_2$, demonstrating the pH dependency of the reaction process. The Michaelis-Menten steady-state kinetics of $GA_{ox}$–Fe was then investigated by using 3,3′,5,5′-tetramethyl-

benzidine (TMB) as the •OH indicator to monitor the time-course reaction process after the addition of different concentrations of $H_2O_2$ (10, 20, 40, and 100 mM) (Fig. 6c). The absorbance changes of the reaction system at 652 nm (characteristic absorption peak for oxidized TMB) can be used to calculate the initial velocities of •OH production according to the Beer-Lambert law, subsequently Michaelis-Menten fitting can be obtained (Fig. 6d). The Michaelis-Menten constant ($K_m$) of $GA_{ox}$–Fe was calculated to be 24.805 mM, which is in between those of horseradish peroxidase (HRP) and $Fe_3O_4$ nanoparticles (Supplementary Table 8)[36], demonstrating its high catalytic activity. The $GA_{ox}$–Fe nanoparticle can be considered as a Fenton-like agent with HRP-mimicking activity[37]. Comparatively, the $K_m$ value of the Fe-HMSN-PEG nanocarrier is much higher at 171.809 mM (Supplementary Fig. 23), demonstrating that the catalytic activity of $GA_{ox}$–Fe is much superior to that of FHPG nanomedicine, and consequently the transformation from the FHPG nanomedicine to iron gall nanocomplex will significantly enhance the catalytic performance toward $H_2O_2$ conversion to •OH.

The high catalytic activity of $GA_{ox}$–Fe may be attributed to the existence of weakened ligand field, which endows the Fe (III) center with insignificant but non-negligible redox property of divalent iron[38,39]. Although the catalytic activity is mainly stemmed from the $Fe^{3+}/Fe^{2+}$ cycling within the nanocomplex, as implied by the results of the second round of CA, however, the weakened ligand–field interaction would largely facilitate the reductive conversion of $Fe^{3+}$ to $Fe^{2+}$, thereby promoting the Fenton-like reaction process. We propose the mechanism of the nanocatalytic •OH-generating reactions triggered by $GA_{ox}$–Fe (Fig. 6e), in which a transient one-electron oxidation product of $GA_{ox}^{2-}$ i.e., $GA_{ox}•^{1-}$, is tentatively assumed for the purpose of simplified presentation of the ligand-metal reduction in the nanocomplex favoring the catalytic process, though the existence of such a product is hard to be confirmed. The metal–ligand cooperativity in $GA_{ox}$–Fe increase the potential of $Fe^{3+}$ reduction to $Fe^{2+}$ during the Fenton-like reactions, which is difficult to achieve by conventional Fenton-like agents made of metal oxides[40].

The capability of two-electron oxidation of pristine GA–Fe, as well as the nanocatalytic property of $GA_{ox}$–Fe toward $H_2O_2$ decomposition to •OH, inspire us to further explore the thorough pro-oxidation pathway of pristine GA–Fe. Distinct •OH signal could be observed in the ESR spectrum of mild acidic buffer solution (pH = 6.5) containing fresh GA–Fe and $O_2$ (Fig. 6f), while no •OH signal was detected in the absence of $O_2$, demonstrating that pristine GA–Fe alone cannot produce •OH but can convert $O_2$ to •OH. Distinct rhodamine B (RhB) decolorization has been observed in buffer solution (pH = 6.5) containing fresh GA–Fe, $O_2$, and RhB (Fig. 6g), while the addition of superoxide dismutase (SOD) or catalase largely compromised the decolorization efficiency, indicating that the $O_2•^-$ and $H_2O_2$ are indispensable intermediates during the conversion of $O_2$ to •OH. Based on these experimental results, we here propose the thorough pro-oxidation mechanism of pristine GA–Fe (Fig. 6h). The $O_2$ first undergoes one-electron reduction by pristine GA–Fe to generate $O_2•^-$, which is then further converted to $H_2O_2$ via a second one-electron reduction reaction. During the two steps of ORRs, the GA–Fe complex serves as a reactant, i.e., reducer, and is resultantly oxidized to $GA_{ox}$–Fe, which then acts as a highly effective Fenton nanocatalyst to catalyze the decomposition of $H_2O_2$ to •OH. The strong metal–ligand exchange coupling between Fe center and gallate ligands endows GA–Fe nanoparticles with high reducibility to promote the consecutive reduction of oxygen species, by which the oxidizing potentials of ROS are significantly elevated from $O_2•^-$ to •OH.

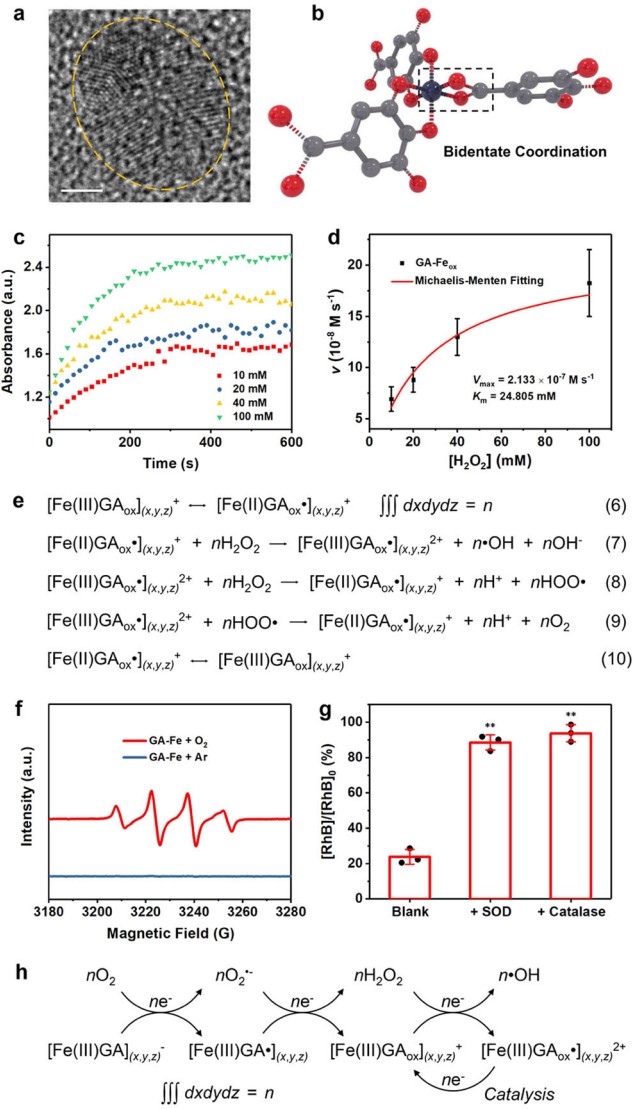

**Fig. 6 Nanocatalytic performance of $GA_{ox}$–Fe and pro-oxidation property of GA-Fe. a** High-resolution TEM image of a single $GA_{ox}$–Fe nanoparticle showing the well-crystallized nature. Scale bar, 2 nm. A representative image of three replicates is shown. **b** Proposed coordination mode of gallate molecules (here should be $GA_{ox}^{2-}$) around each Fe center in $GA_{ox}$–Fe. The connection between the carboxylic anion of gallate and Fe center has shifted from bridge-type coordination to bidentate coordination. **c** Time-course absorbance of buffer solution (pH = 6.5) containing $GA_{ox}$–Fe and TMB after adding different concentrations of $H_2O_2$. **d** Michaelis-Menten kinetics of $GA_{ox}$–Fe based on (**c**). For one specific $H_2O_2$ concentration, the initial velocity of catalytic reaction was calculated by averaging the mean velocities of initial eight periods in (**c**) (15 s per period). Data are expressed as mean ± SD ($N = 8$ independent experiments). **e** Proposed nanocatalytic reactions triggered by $GA_{ox}$–Fe mimicking the property of HRP. The $H_2O_2$ generated from ORRs enabled via the oxidation of fresh GA–Fe in Fig. 4e can be further catalytically converted to •OH by $GA_{ox}$–Fe as a Fenton agent. **f** ESR spectra evaluating •OH generation in buffer solution (pH = 6.5) containing fresh GA–Fe under the addition of $O_2$ or Ar. **g** RhB decolorization in buffer solution (pH = 6.5) containing fresh GA–Fe with the presence of $O_2$ (blank group). SOD or catalase has also been added for investigating the generation of $O_2^{•-}$ and $H_2O_2$ intermediates during the pro-oxidation reactions, respectively. Data are expressed as mean ± SD ($N = 3$ independent experiments). **P < 0.01, based on the Student's two-sided t-test. **h** Proposed pro-oxidation mechanism of GA-Fe. The nano-metalchelate serve as a reactant in the former two one-electron reaction steps to successively reduce oxygen to $O_2^{•-}$ and then to $H_2O_2$, after which it acts as a nanocatalyst to catalyze the decomposition of $H_2O_2$ to •OH. It is noted that additional $n$ mol of $H_2O_2$ is required to reduce the $[Fe(III)GA_{ox}•]_{(x,y,z)}^{2+}$ intermediate in the third and fourth steps of (**e**) for sustaining the catalytic process. It can also be considered that only half-amount of generated $H_2O_2$ has been catalytically converted to •OH while the other half was used for reducing $[Fe(III)GA_{ox}•]_{(x,y,z)}^{2+}$, dependent on the specific chemical environments. However, here we only present the first scenario for a simplified illustration. Source data are provided as a Source Data file.

can be internalized into cells within 2 h of incubation (Supplementary Fig. 25). Intriguingly, bio-TEM image of HeLa cells after FHPG incubation for 6 h reveals the formation of numbers of heterogeneous solid nanoparticles with a high image contrast and a high crystallinity after partial FHPG degradation (Fig. 7b and Supplementary Fig. 26), evidencing the feasibility for the transformation of FHPG to GA–Fe in acidic cancer cellular environment, thanks to the intrinsic acidity-triggered degradability of the Si–O–Fe hybrid framework of nanocarrier, as well as the high rate of the coordination reaction between $Fe^{3+}$ and gallate. Comparatively, no distinct morphological transformation of FHPG occurred in HUVECs in the same period of nanomedicine incubation (Fig. 7c), and no heterogeneous solid nanoparticles formed, confirming the cancer specificity of in situ GA–Fe synthesis from FHPG.

To investigate whether the pro-oxidation reactions take place in cancer cells after FHPG incubation or not, the •OH indicator 2′, 7′-dichlorofluorescein diacetate (DCFH-DA) has been used to treat cancer cells, which can be oxidized to 2′, 7′-dichlorofluorescein (DCF) emitting green fluorescence[41]. Flow cytometry indicates a distinct green fluorescence signal in HeLa cells after FHPG treatment for 6 h (Fig. 7d and Supplementary Fig. 27), manifesting the generation of a large amount of •OH. Comparatively, no distinct green fluorescence signal was detected in HeLa cells of Fe-HMSN-PEG and gallate groups, demonstrating that the formation of GA–Fe complex from FHPG is the prerequisite to trigger pro-oxidation reactions. The generation of a minor amount of •OH in Fe-HMSN-PEG group may result from the catalytic conversion of endogenous $H_2O_2$ to •OH, which

Comparatively, ESR spectra indicate no •OH signal in mild acidic buffer solution (pH = 6.5) containing Fe-HMSN-PEG under the presence of $O_2^{•-}$ or $O_2$ (Supplementary Fig. 24), demonstrating that the nanocarrier of FHPG nanomedicine is not capable of triggering ORRs directly, presenting a relative redox inertness compared with the GA–Fe coordination complex. Therefore, the acidity-triggered degradation of FHPG and the in situ synthesis of highly reductive iron gall nanocomplex will favor the occurrence of ORRs preferentially in acidic environment. This chemical characteristic of FHPG is beneficial for subsequent cancer therapeutic application by generating highly oxidizing •OH specifically in acidic tumor region.

**Cancer-specific GA–Fe formation and ROS production.** Based on the above experimental explorations on the chemical properties of FHPG and GA–Fe, it is here expected that the acidic intracellular environment of cancer cells will promote the degradation-enabled evolution of FHPG to GA–Fe nanocomplex favoring subsequent ORRs and •OH generation, finally triggering oxidative damage to cancer cells (Fig. 7a). The anticancer function of FHPG was first investigated by incubating the nanomedicine with human cervical cancer cell line HeLa and human umbilical vein endothelial cells (HUVECs). Confocal laser scanning microscopy (CLSM) images indicate that the nanomedicine

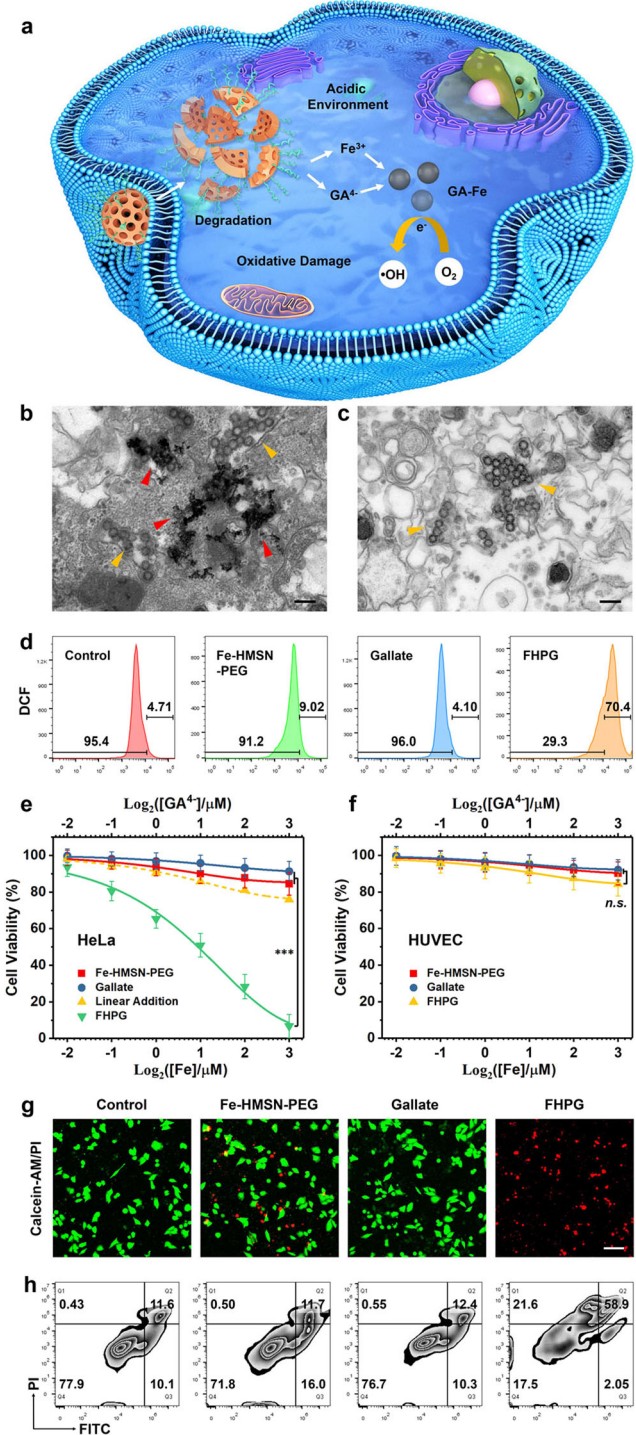

**Fig. 7 Cancer-specific synthesis of nano-metalchelate triggering distinct oxidative damage in cancer cells. a** Scheme for the anticancer mechanism of FHPG by acidity-responsive degradation and subsequent formation of nano-metalchelate enabling pro-oxidation reactions in cancer cells. **b**, **c** Bio-TEM images of HeLa cells (**b**) and HUVECs (**c**) after treatment with FHPG for 6 h. Scale bars, 500 nm. A representative image of three replicates from each group is shown. Yellow triangle marks indicate FHPG while red ones indicate the GA–Fe nano-metalchelate newly formed in the cancer cells after FHPG degradation. **d** Flow cytometry investigating •OH generation in HeLa cells after various treatments for 6 h, representative of 3 independent experiments. **e**, **f** Relative viabilities of HeLa cells (**e**) and HUVECs (**f**) after different treatments for 24 h. Linear addition for the effect of single Fe-HMSN-PEG or gallate treatment on the relative viability of HeLa cells is plotted for comparison with that of FHPG group. Data are expressed as mean ± SD ($N = 6$ independent experiments). $^{***}P < 0.001$, n.s., not significant, based on the Student's two-sided $t$-test. **g** CLSM images of HeLa cells after different treatments for 24 h. Calcein-AM and PI were used for cell alive/dead observation. Scale bar, 100 μm. A representative image of three replicates from each group is shown. **h** Flow cytometry evaluating the death mechanism of HeLa cells after indicated treatments for 24 h, representative of 3 independent experiments. Annexin V-FITC and PI were used to stain cells for differentiating their living states. Source data are provided as a Source Data file.

generate $O_2^{\bullet-}$ and $H_2O_2$ intermediates intracellularly, favoring subsequent •OH generation by providing the reaction substrate.

The anticancer efficiency and specificity of FHPG were then investigated by using cell counting kit-8 (CCK-8) assay and CLSM to quantify or visualize the viability of HeLa cells and HUVECs after incubation with different concentrations of FHPG for 24 h. Fe-HMSN-PEG and gallate groups were also set for comparison. In addition, murine breast cancer cell line 4T1 and murine breast epithelial cell line HC11 were also used for a comprehensive evaluation of the anticancer effect of these treatments. Fe-HMSN-PEG or gallate alone could not trigger a distinct reduction of viability of cancer cell lines (HeLa and 4T1) (Fig. 7e, g and Supplementary Fig. 29a), while the FHPG nanomedicine shows a significant anticancer effect, much more distinct than the theoretical addition of those of single Fe-HMSN-PEG and gallate treatments, attributing to the acidity-triggered nanomedicine degradation and GA–Fe nanocomplex formation that promote the generation of large amount of •OH and finally trigger distinct oxidative damage in cancer cells, which is not achievable by single Fe-HMSN-PEG or gallate treatment alone. The synergy between the nanocarrier and the loaded gallate in one nanosystem enables the coordination reaction between released GA and Fe ions, and the subsequent formation of GA–Fe coordination complex in cancer cells, underpinning subsequent significant ROS production in cancer cells benefiting from the strong metal–ligand cooperativity of GA–Fe. Comparatively, in normal cell lines (HUVEC and HC11) the FHPG nanomedicine could not trigger a distinct cell death (Fig. 7f and Supplementary Fig. 29b), due to the pH responsiveness of the –Si–O–Fe– framework of FHPG that makes the nanocarrier uncapable of degrading in neutral intracellular environment of normal cells, thereby guaranteeing the biosafety of the nanomedicine and its anticancer specificity.

The cell death mechanism was further investigated via flow cytometry. Both HeLa cells and HUVECs were stained with annexin V-fluorescein isothiocyanate (FITC) and propidium iodide (PI) after different treatments for flow cytometric analysis and comparison. Distinct enhancements of late apoptosis signal (Q2 quadrant) and necrosis signal (Q1 quadrant) have been confirmed in HeLa cells after FHPG treatment for 24 h (Fig. 7h

is apparently not sufficient compared with GA–Fe-triggered pro-oxidation reactions. To further confirm the occurrence of two steps of sequential one-electron ORRs in cancer cells triggered by the formed GA–Fe nanocomplex, primary HeLa cells were also infected with adenovirus encoding SOD or catalase before FHPG treatment to scavenge the intracellularly generated $O_2^{\bullet-}$ or $H_2O_2$, respectively, then the cells after FHPG treatment were stained with DCFH-DA and analyzed via flow cytometry (Supplementary Fig. 28). Much weakened DCF signal could be detected in HeLa cells after infection by adenovirus encoding either SOD or catalase, demonstrating that the GA–Fe nanocomplex can trigger two steps of sequential one-electron ORRs in cancer cells to

and Supplementary Fig. 30), while the single nanocarrier or gallate treatment could not trigger significant changes of apoptosis or necrosis signals in HeLa cells, evidencing that the synergistic actions of Fe-engineered nanocarrier and the loaded gallate in one nanosystem is the prerequisite for triggering distinct anticancer effect. In addition, the insignificant changes of apoptosis or necrosis signals could be observed in HUVECs after FHPG treatment for 24 h (Supplementary Figs. 31 and 32), as FHPG degrades preferentially in acidic environments, while in neutral intracellular environment of normal cells the nanomedicine is stable enough to prevent GA–Fe nanocomplex generation, not to mention subsequent series of ORR and Fenton reactions.

**In vivo antitumor efficacy and biosafety.** The high and specific cytotoxicity of FHPG demonstrated in cellular experiments encouraged us to evaluate in vivo anticancer efficacy on animal models. Xenografted HeLa cervical cancer model was established by injecting HeLa cells into the thigh of female Balb/c mice (four-week-old). When the tumor volume reached to the dimension of about 100 mm³, a part of mice bearing HeLa tumor xenografts was selected for pharmacokinetic evaluations of FHPG nanomedicine via intravenous injection. However, given the erythrocyte in blood flow are rich in Fe-containing hemoglobin, here we selected Si element instead of Fe for investigating the hemodynamics of FHPG and its time-dependent distribution in the main organs of the body (hearts, livers, spleens, lungs, and kidneys) and tumors after blood circulation. The blood circulation half-time ($T_{1/2}$) of FHPG was calculated to be approximately 1.75 h (Fig. 8a) according to a two-compartment model, which is roughly equivalent to that of PEGylated pristine MSNs (1.85 h) based on our previous report[42]. The systemic distribution of the Si element also reveals a 1.12% ID g⁻¹ of tumor passive accumulation efficiency (Fig. 8b), higher than the median value for tumor accumulation efficiencies of current reported cancer nanomedicines (0.7% ID g⁻¹)[43]. Additionally, metabolic profiles also reveal the easy excretion of FHPG out of body via urine and feces (Supplementary Fig. 33), suggesting the biocompatibility of the nanomedicine.

Twenty HeLa-tumor-bearing mice were randomly divided into 4 experimental groups ($N = 5$). The antineoplastic efficacy evaluation of FHPG was then conducted by the intravenous injection of PBS solution of nanomedicine to one group of tumor-bearing mice followed by 2 weeks of observation (Fig. 8c). The other 3 groups of mice received the treatments with PBS or PBS containing single Fe-HMSN-PEG or gallate, respectively, for comparison. Significant tumor growth inhibition could be observed in mice after FHPG administration for 14 days (Fig. 8d), while single nanocarrier or gallate treatment led to negligible tumor suppression. This distinct difference of antitumor efficacy between FHPG nanomedicine and single Fe-HMSN-PEG/gallate treatment is believed to result from the generation of GA–Fe complex from FHPG in cancer cells that leads to the distinct oxidative damage against tumors. Mice in the 4 experimental groups show no distinct body weight fluctuations (Fig. 8e), as a consequence of their relative high biocompatibility. A prolonged lifespan of mice has been visualized in FHPG group compared with other 3 groups, according to the Kaplan-Meier survival curves (Supplementary Fig. 34), which may be owing to the remarkable suppression on the growth of HeLa tumor xenografts after FHPG treatment. Additionally, histochemical assay for the expression of GPX4, a typical indicator for oxidative stress, indicates that the FHPG nanomedicine could efficiently trigger oxidative damage to tumors (Fig. 8f). Hematoxylin and eosin

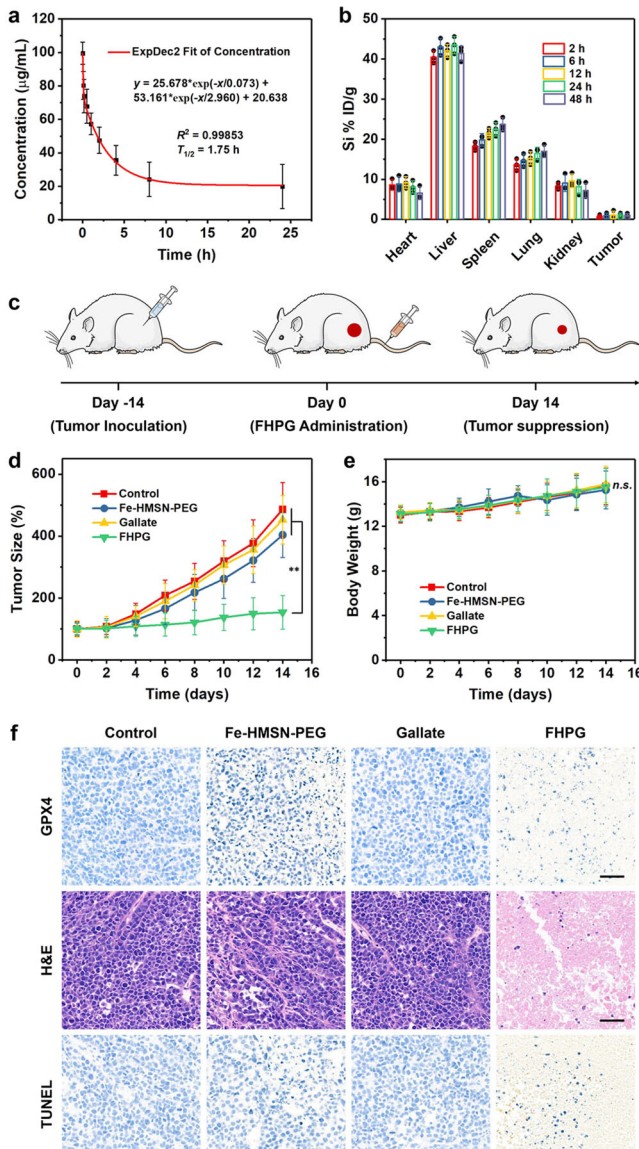

**Fig. 8 Anticancer efficacy evaluation in vivo. a** Time-dependent concentration of FHPG in blood of HeLa-tumor-bearing mice after intravenous administration. Data are expressed as mean ± SD ($N = 3$ biologically independent animals). The circulation half-time ($T_{1/2}$) of FHPG was calculated to be 1.75 h. **b** Distribution of Si element in main organs (hearts, livers, spleens, lungs, and kidneys) and tumors of mice at varied time points after FHPG administration. Data are expressed as mean ± SD ($N = 3$ biologically independent animals). **c** Scheme for the schedule of in vivo FHPG treatment. **d** Time-dependent growth curves of HeLa tumor xenografts in mice after treatments with: (1) PBS (control), (2) Fe-HMSN-PEG, (3) gallate, or (4) FHPG. Data are expressed as mean ± SD ($N = 5$ biologically independent animals). **P < 0.01, based on the Student's two-sided t-test. **e** Body weight changes of HeLa-tumor-bearing mice after various treatments in (**d**). Data are expressed as mean ± SD ($N = 5$ biologically independent animals). n.s., not significant, based on the Student's two-sided t-test. **f** Histochemical analyses (GPX4, H&E, TUNEL) of HeLa tumor tissue harvested from mice after various treatments on day 14, representative of five biological replicates from each experimental group. Scale bar, 50 μm. Source data are provided as a Source Data file.

(H&E) staining and terminal deoxynucleotidyl-mediated dUTP nick-end labeling (TUNEL) staining of tumor tissue sections reveal the distinct apoptosis of cancer cells after FHPG treatment, further evidencing the potent antitumor effect of FHPG nanomedicine.

As the FHPG nanomedicine was administrated intravenously, a portion of nanomedicines will enter normal organs. Although the nanosystem degrade preferentially in acidic tumor region, however, the possible nanomedicine degradation in neutral environment of normal organs cannot be totally excluded. Therefore, the in vivo biosafety of FHPG, such as hematological and histological safety, should be systematically investigated. The key hematological, hepatic, and renal parameters of HeLa-tumor-bearing mice after FHPG treatment show insignificant differences from those of mice after other treatments (Supplementary Fig. 35), demonstrating the satisfactory biocompatibility of the nanomedicine. Additionally, H&E staining indicates no distinct pathological changes in major organs (hearts, livers, spleens, lungs, and kidneys) of tumor-bearing mice after different treatments for 14 days (Supplementary Fig. 36), while the GPX4 expressions in major organs of mice in all experimental groups show normal expression levels (Supplementary Fig. 37), demonstrating the excellent compatibility and negligible side effect of FHPG.

## Discussion

The iron gall ink-triggered chemical corrosion of the historical documents in Western cultural heritages has inspired us to explore the metal–ligand coordination effects in iron gall complex underlying ORR and ROS generation, and to apply such a pro-oxidative mechanism of GA–Fe to biomedical applications, such as anticancer therapy. In this work, we report a composite nanomedicine FHPG for triggering oxidative damage in tumors, which was constructed by loading gallate, also a key bioactive component of tea leaves with high biocompatibility, into an Fe-engineered nanocarrier. The –Si–O–Fe– hybrid framework of the nanocarrier presents acidity responsiveness that can degrade selectively in acidic environment, after which $Fe^{3+}$ and the loaded gallate are co-released, triggering coordination reactions between the two chemicals to generate GA–Fe nano-chelate. The hex-acoordinated iron coordination compound shows a strong metal–ligand exchange coupling between Fe center and gallate ligands, endowing the nano-chelate with high reducibility to promote two sequential one-electron ORRs generating $O_2^{\bullet-}$ and then $H_2O_2$. Moreover, the two-electron oxidation form of the GA–Fe ($GA_{ox}$–Fe) with a remaining but weakened ligand field is also an excellent Fenton-like agent that can catalyze the decomposition of generated $H_2O_2$ into highly oxidizing •OH, finally triggering severe oxidative damage to tumors. Cellular experiments evidence the cancer-specific synthesis of the nano-metal-chelate, which presents high anticancer effect. Animal experiments further reveal the high therapeutic efficacy of FHPG nanomedicine and its desired biosafety. It is expected that such a therapeutic approach is instructive for further cancer therapy.

## Methods

**Chemicals and reagents.** CTAC, TEA, xanthine, and xanthine oxidase (XO) were purchased from Sigma-Aldrich. TEOS and NaOH were obtained from Shanghai Lingfeng Chemical Reagent Co., Ltd. Fe(acac)₃ was provided by J&K Scientific. Urea, RhB, iron chloride hexahydrate (FeCl₃·6H₂O) and 3,3′,5,5′-tetramethylbenzidine (TMB) were bought from Sinopharm Chemical Reagents Co., Ltd. Methoxy PEG silane was purchased from Shanghai Yare Biotech, Co., Ltd. Gallic acid was acquired from Macklin. Dulbecco's modified eagle's medium (DMEM), fetal bovine serum (FBS), penicillin/streptomycin, and PBS were provided by Gibco. FITC, 4′, 6-diamidino-2-phenylindole (DAPI), DCFH-DA, CCK-8 assay, calcein-AM, PI, annexin V-FITC, H&E, paraformaldehyde (PFA) and TUNEL apoptosis assay kit were acquired from Beyotime. Primary antibody against GPX4

and a biotinylated secondary antibody (anti-rabbit IgG, HRP-linked antibody) were purchased from Cell Signaling Technology.

**Synthesis of MSNs.** CTAC (2 g) and TEA (0.02 g) were homogenized into deionized water (18 mL, 80 °C) by magnetic stirring for 30 min, followed by the addition of TEOS (1.5 mL) dropwise. After reaction for 4 h, the suspension was centrifuged and the obtained precipitate was further washed with water and ethanol three times, respectively, then redispersed into an HCl–ethanol mixed solution (10%, 200 mL) overnight for removing surfactant. Finally, the suspension was centrifuged and the precipitate was rinsed with water and ethanol three times to obtain MSNs.

**Preparation of Fe-HMSNs.** MSN (25 mg), urea (1.35 g) and Fe(acac)₃ (88 mg) were homogenized in a water–ethanol mixed solution (water: 7.5 mL, ethanol: 5 mL), then the mixture was allowed to react at 200 °C for 48 h. After cooling down to room temperature, the obtained product was rinsed with water and ethanol three times, respectively.

**PEGylation.** Fe-HMSNs (100 mg) and methoxy PEG silane (50 mg) were homogenized in ethanol (100 mL), followed by heating at 78 °C to allow the reaction for 24 h. After cooling down to room temperature, the obtained product was rinsed with water and ethanol three times, respectively.

**Preparation of FHPG.** Gallate solution was first prepared by dispersing gallic acid (70 mg) into deionized water (50 mL) at room temperature, then the pH value of the solution was regulated to 7.4. Fe-HMSNs-PEG (150 mg) were dispersed into the solution under magnetic stirring for 1 h, after that the suspension was centrifuged and the precipitate was rinsed with water three times to obtain FHPG. The mass ratios of Fe and gallate in FHPG are 5.707 and 16.817 wt%, respectively, for keeping the stoichiometric ratio of Fe to gallate to be 1:1.

**Characterization.** TEM images, SAED patterns, HAADF image, element mappings, and EDS profile were obtained on JEM-2100F microscope (JEOL). SEM images were acquired on SU9000 (HITACHI). Nitrogen adsorption–desorption isotherm and pore-size distribution data were obtained on Quadrasorb SI (Quantachrome). XRD patterns were acquired on Ultima IV X-ray diffractometer (Rigaku). ²⁹Si solid-state MAS-NMR spectra were acquired on AVANCE III HD solid NMR spectrometer (Bruker). XPS spectra were recorded by ESCALab250 (Thermal Scientific). FTIR spectra were acquired on Nicolet iS 10 (Thermo Scientific). The gallate concentration in the solution was determined by observing the peak value of UV-Vis absorption spectra at 214 or 263 nm on UV-3600 spectrometer (Shimadzu). The Fe and Si concentrations in the solution were determined by inductively coupled plasma optical emission spectrometer (ICP-OES) (Agilent Technologies). Raman spectra were acquired on inVia variable temperature Raman spectrometer (Renishaw). CV and CA were conducted on CHI 760E electrochemical workstation (CH Instrument). ESR spectra were acquired on EMX plus spectrometer (Bruker). CLSM images were obtained on FV1000 (Olympus). Flow cytometry was conducted on LSRFortessa (Becton Dickinson).

**Degradation of the nanocarriers.** Fe-HMSNs-PEG (40 mg) was dispersed in SBF (40 mL, pH = 6.5 or 7.4) under magnetic stirring. An aliquot (1 mL) was extracted from the solution at different time intervals (2, 4, 6, 8, 10, 12, 24, 36, 48, and 60 h) and centrifuged. The supernatant was used for determining the concentrations of released Fe and Si elements by ICP-OES, while the precipitate was redispersed in ethanol for further observation under TEM.

**Nano-metalchelate formation.** FHPG (40 mg) was dispersed in SBF (40 mL, pH = 6.5 or 7.4, respectively). After 60 or 90 h of magnetic stirring, an aliquot (1 mL) was extracted from the solution for TEM observation of the generated GA–Fe nano-metalchelate. Given the oxidation of GA–Fe nanoparticles will inevitably occur during FHPG degradation, fresh GA–Fe was also prepared additionally for precisely evaluating the redox characteristics of the nano-metalchelate. FeCl₃·6H₂O was added into the gallate solution to keep a 1:1 stoichiometry of $Fe^{3+}$ and $GA^{4-}$ for reaction with magnetic stirring and nitrogen supplementation for 1 h, after which the suspension was centrifuged and the precipitate was washed, collected, and stored for further use.

**Electrochemical oxidation.** A three-electrode cell was applied for performing CV and CA with a nickel foam electrode, an Ag/AgCl electrode (saturated with KCl), and a carbon rod electrode as the working electrode, reference electrode, and auxiliary electrode, respectively.

*CV.* CV curve were recorded between −0.50 and 0.90 V (scan rate: 50 mV s⁻¹) in a Na₂SO₄ aqueous solution (1 M, pH = 6.5) containing fresh GA–Fe ([Fe] = [$GA^{4-}$] = 15 mM). The electrolyte solution was purged with nitrogen for 20 min before starting CV measurement. The CV curves of electrolyte solutions containing single $Fe^{3+}$ (15 mM) or $GA^{4-}$ (15 mM) have also been recorded.

*CA*. CA was firstly conducted with two step potentials (high potential: 0.90 V; low potential: 0.15 V; step frequency: $0.0187 \, s^{-1}$; period: 107 s) in a $Na_2SO_4$ aqueous solution (1 M, pH = 6.5) containing $Fe^{3+}$ (15 mM) or fresh GA–Fe ([Fe] = [$GA^{4-}$] = 15 mM). After 7 periods of oxidation, the GA–Fe-containing electrolyte solution was centrifuged and the supernatant was used for measuring the concentrations of released Fe elements by ICP-OES, while the precipitate was redispersed in a fresh $Na_2SO_4$ aqueous solution (1 M, pH = 6.5) for a second round of DPSCA measurement (high potential: 0.90 V; low potential: −0.45 V; step frequency: $0.0187 \, s^{-1}$; period: 107 s). After 7 periods of oxidation and reduction, the electrolyte solution was centrifuged and the supernatant was used for measuring the concentrations of released Fe elements by ICP-OES, while the precipitate was rinsed with water three times and collected for subsequent characterizations.

## Catalytic performance

*ESR measurement*. The collected precipitate, i.e., electrochemically oxidized GA–Fe, was redispersed in MES buffer solution (pH = 6.5, 7.0, or 7.4, [Fe] = 4 mM for the nanoparticles) supplemented with $H_2O_2$ (50 μM). After 10 s of reaction, DMPO was added into the system and an aliquot of the mixture was extracted for ESR measurement.

*Michaelis-Menten kinetics*. The electrochemically oxidized GA–Fe was redispersed in MES buffer solution (pH = 6.5, [Fe] = 10 μM for the nanoparticles), followed by the supplementation with $H_2O_2$ (10, 20, 40, and 100 mM) and TMB (0.8 mg $mL^{-1}$). The variations of the absorbances at 652 nm of the solutions were monitored using a SpectraMax M2 microplate reader in a kinetic mode. The absorption values could be converted into the concentration of oxidized TMB in the mixture via Beer-Lambert law:

$$A = \varepsilon b c \tag{1}$$

where $A$ is the absorption value of the solution at 652 nm, $\varepsilon$ is a constant for the molar absorption coefficient of oxidized TMB: 39,000 $M^{-1}cm^{-1}$, $b$ is the optical length of the solution, and $c$ is the concentration of oxidized TMB in the reaction system. The Michaelis-Menten kinetics of electrochemically oxidized GA–Fe were evaluated by plotting the initial reaction velocities against $H_2O_2$ concentrations according to the following equation:

$$v_o = \frac{V_{max} \times [H_2O_2]}{K_M + [H_2O_2]} \tag{2}$$

where $v_o$ is the initial reaction velocity, $V_{max}$ is the maximum reaction velocity, $[H_2O_2]$ is the $H_2O_2$ concentration, and $K_M$ is the Michaelis-Menten constant. The values of $V_{max}$ and $K_M$ were calculated through the software Origin Pro (version 2017). The Michaelis-Menten kinetics of Fe-HMSN-PEG have also been evaluated for comparison.

## Redox-regulating mechanism

*ESR measurement*. Fresh GA–Fe was redispersed in MES buffer solution (pH = 6.5, [Fe] = 4 mM for the nanoparticles) supplemented with oxygen or argon. After reaction for 30 min, DMPO was added into the system and an aliquot of the mixture was extracted for ESR measurement.

*RhB decolorization*. Fresh GA–Fe was redispersed in MES buffer solution (pH = 6.5, [Fe] = 4 mM for the nanoparticles) containing RhB (0.01 mM) supplemented with oxygen. After 6 h of reaction, the oxidation of RhB was measured by observing the values of characteristic peak at 554 nm on UV-Vis absorption spectra. SOD or catalase was also added to confirm the generation of $O_2^{\bullet-}$ and $H_2O_2$ intermediates during reactions.

## Cell culture

Human cervical cancer cell line HeLa, human umbilical vein endothelial cells (HUVECs), murine breast cancer cell line 4T1, and murine breast epithelial cell line HC11 were kindly provided by Cell Bank/Stem Cell Bank, Chinese Academy of Sciences. These cell lines were cultured on 75 $cm^2$ cell culture flasks containing DMEM with the addition of 10% FBS and 1% penicillin/streptomycin.

## Cancer-specific nano-metalchelate formation

*CLSM*. HeLa cells were treated with FITC-labeled FHPG for 2 h and observed under FV1000 confocal fluorescence microscope for monitoring the cellular uptake of nanoparticles. DAPI was used for staining nuclei.

*Bio-TEM*. HeLa cells and HUVECs were incubated in 10 cm plates overnight, then treated with FHPG for another 24 h. After that, these cells were collected and fixed to make ultrathin sections for observation under JEM-2100F electron microscope.

## Cellar catalytic •OH generation

HeLa cells and HUVECs were seeded in 6-well plates. After indicated treatments, cells were treated with culture medium containing DCFH-DA. Then cells were rinsed with PBS three times and analyzed using a flow cytometer. To further investigate the cellular pro-oxidation process triggered by GA–Fe generation after FHPG treatment, primary HeLa cells were also infected by adenovirus encoding SOD or catalase before the FHPG treatment to scavenge

the intracellularly generated $O_2^{\bullet-}$ or $H_2O_2$, respectively, then the cells after FHPG treatment were stained with DCFH-DA and analyzed using a flow cytometer.

## Cancer cell death mechanism

*Cell viability assay*. HeLa cells, HUVECs, 4T1 cells, and HC11 cells were seeded in 96-well plates overnight (initial density: $1 \times 10^4$ cells per well), then treated with Fe-HMSN-PEG or gallate or FHPG, dispersed in DMEM for 24 h. After that, the CCK-8 assay was applied to test cell viabilities in each experiment group. The linear addition of the effects of single Fe-HMSN-PEG and gallate treatment on viability of cancer cell lines was calculated according to the following equation:

$$R_{Linear\ addition,x} = 1 - \left[ \left( 1 - \overline{R_{Fe-HMSN-PEG,x}} \right) + \left( 1 - \overline{R_{Gallate,x}} \right) \right] \tag{3}$$

where $R_{Linear\ addition,x}$ indicates the linear addition of the effects of Fe-HMSN-PEG ([Fe] = $x$ μM) and gallate ([$GA^{4-}$] = $x$ μM) on cell viability, $\overline{R_{Fe-HMSN-PEG,x}}$ is the average value of cell viabilities after single Fe-HMSN-PEG treatment ([Fe] = $x$ μM), $\overline{R_{Gallate,x}}$ is the average value of cell viabilities after single gallate treatment ([$GA^{4-}$] = $x$ μM).

*CLSM*. HeLa cells were seeded in 15 mm cell culture dishes overnight, then cells were treated with Fe-HMSN-PEG ([Fe] = 8 μM) or gallate ([$GA^{4-}$] = 8 μM) or FHPG ([Fe] = [$GA^{4-}$] = 8 μM), dispersed in DMEM for 24 h followed with calcein AM and PI staining for alive/dead observation.

*Flow cytometry*. HeLa cells and HUVECs were incubated in 6-well plates. After different treatments, cells were washed and resuspended in a binding solution (0.5 mL), followed by addition of annexin V-FITC and PI. Then the cells were rinsed with PBS three times and analyzed using a flow cytometer. The data were collected via the software CytExpert (version 2.2) and then analyzed via FlowJo (version 10.0).

## Xenograft tumor model

All animal experiment procedures follow the guidelines of the Animal Care Ethics Commission of Shanghai Tenth People's Hospital, Tongji University School of Medicine (ID: SHDSYY-2018-Z0026). Forty 4-week-old female Balb/c mice were purchased from Vital River Laboratories. These mice were housed in ventilated stainless-steel cages under standard conditions (light: 12 h light/dark cycle, ambient temperature: 25 ± 2 °C, humidity: 60 ± 10%), which were fed with pellet food ad libitum and sterilized water. Then, $2 \times 10^6$ HeLa cells were dispersed in PBS (100 μL) and injected into the thigh of mice subcutaneously for establishing HeLa tumor xenograft. The length and width of HeLa tumor xenografts were measured every 2 days and the tumor volumes were calculated according to the following equation:

$$V_x = \frac{L_x \times W_x^2}{2} \tag{4}$$

where $V_x$, $L_x$, and $W_x$ indicate the volume, length, and width of xenografted HeLa tumors after different treatments at day $x$.

## Pharmacokinetics investigation

*Blood circulation lifetime*. Mice with HeLa tumor xenografts were injected with PBS dispersing FHPG (20 mg $kg^{-1}$) intravenously. At given time points (2, 5, and 10 min, 0.5, 1, 2, 4, 8, and 24 h), an aliquot of blood (15 μL) was collected and the Si element concentration was measured by ICP-OES. The blood terminal half-life of FHPG was calculated by a double component model.

*Biodistribution*. At given time points after FHPG administration (6, 12, 24, and 48 h), the major organs (hearts, livers, spleens, lungs, and kidneys) and tumors of mice were harvested and then treated with aqua regia for dissolution. The Si element contents were measured by ICP-OES.

*Metabolism study*. The urine and feces of HeLa-tumor-bearing mice were collected at different time points after FHPG administration (2, 6, 12, 24, and 48 h), then treated with aqua regia for dissolution. The contents of Fe and Si in urine and feces were measured by ICP-OES.

## Antineoplastic effects

*In vivo chemotherapy*. When the volume of HeLa tumor xenografts reached approximately 100 $mm^3$, 20 tumor-bearing mice were divided into 4 groups ($N = 5$) randomly and administered intravenously with PBS (50 μL) or equal volume of PBS containing Fe-HMSN-PEG (20 mg $kg^{-1}$) or gallate (20 mg $kg^{-1}$) or FHPG (20 mg $kg^{-1}$), respectively. The tumor volumes, survival times, and body weights of mice were recorded every 2 days.

*Histology*. Experimental mice after different treatments were euthanized at day 14, then the xenografted HeLa tumor tissues were obtained and immersed in 4% PFA for fixation. Tumor sections were treated with H&E or TUNEL for observation under light microscopy. Additionally, tumor sections were also treated with primary antibody against GPX4 (1:1000 dilution), then incubated with a biotinylated

secondary antibody (anti-rabbit IgG, HRP-linked antibody, 1:2000 dilution). An ABC peroxidase standard staining kit was used for determining the content of the antibody complex.

**Biosafety**

*Hematology*. Blood samples (0.8 mL) were collected from HeLa-tumor-bearing mice in different experimental groups after eyeball extraction on day 28, then the blood biochemistry assays were conducted at Shanghai Research Center for a Biomodel Organism.

*Histology*. The hearts, livers, spleens, lungs, and kidneys were collected from HeLa-tumor-bearing mice in different groups on day 14, followed by 4% PFA treatment. H&E staining was applied to stain these tissue sections. Additionally, these tissue sections were also treated with primary antibody against GPX4 and incubated with a biotinylated secondary antibody subsequently. An ABC peroxidase standard staining kit was used for determining the content of the antibody complex.

**Statistical analysis**. Data for $n \geq 3$ independent experiments were expressed as mean ± standard deviation (SD). The statistical significances in this work were analyzed via a two-sided Student's $t$-test using the software SPSS 20 statistics (version 26.0), n.s., not significant; $^*P < 0.05$, significant; $^{**}P < 0.01$, moderately significant; $^{***}P < 0.001$, highly significant.

**Reporting summary**. Further information on research design is available in the Nature Research Reporting Summary linked to this article.

## Data availability

The authors declare that all data needed to support the finding of this study are presented in the article and the Supplementary information. Any data related to this work are available from the corresponding authors upon reasonable request. A reporting summary for this article is available as a Supplementary Information file. Source data are provided with this paper.

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

## Acknowledgements

We greatly acknowledge the financial support from the National Natural Science Foundation of China (Grant No. 21835007), Key Research Program of Frontier Sciences, Chinese Academy of Sciences (Grant No. ZDBS-LY-SLH029) and Shanghai Municipal Government S&T Project (Grant No. 17JC1404701).

## Author contributions

B.Y. and J.S. designated the idea of this work. B.Y., H.Y., H.T., Z.Y., Y.G., Y.W., J.Y., and C.C. synthesized the nanomedicine and performed in vitro and in vivo experiments. B.Y. wrote the whole manuscript. J.S. supervised the project, revised the manuscript, and commented on it.

## Competing interests

The authors declare no competing interests.
