## [Peer Review File. · Nature Communications]

Reviewers' Comments:

Reviewer #1:

Remarks to the Author:

The authors describe the synthesis of a GA/Fe Fe-engineered mesoporous silica nanocarrier for the treatment of cancer. The authors provide comprehensive materials characterisation using multiple experiential approaches.

They draw upon an well know chemical reaction that is attributed to the breakdown of written works. Their approach to use the underlying chemistry responsible for this to treat cancer tumours is very novel. The authors provide clear and scientifically sound discussions and conclusions based on their experimental data at all stage of materials development and cellular evaluation.

Please provide some clarification in the text regarding the possible dangers of the mesoporous material degrading at neutral pH. Whilst this degradation is shown to be slow, what is the likely hood that the nanocarrier could breakdown outside the tumour environment and cause damage to other organs. The research does not use a tumour targeted approach and relies on the vascular system to deliver the material to the tumour, therefore off target effects would be likely.

Many of the figures contain data with error bars. Are these standard errors, standard deviations or something else? Also there was no statistics provided on the significance of data variation.

The level of experimental detail was sufficient for a researcher to reproduce the work.

This paper provides novel method to cancer tumour treatment and i feel the work will be of interest to researchers and clinicians working in the oncology field. I also feel the comprehensive materials synthesis and characterisation will be of much interest to a wide materials research community.

Reviewer #2:

Remarks to the Author:

In this study, the authors prepared a nanosystem for cancer therapy by loading gallate into a Fe-engineered hollow silica nanocarrier. Intracellularly released Fe³⁺ and gallate can trigger the coordination reactions with each other to form GA-Fe nanocomplex in situ, leading to oxygen reduction reaction for generating hydrogen peroxide. In addition, GA-Fe as a Fenton-like agent can catalyze hydrogen peroxide into hydroxyl radical to provide oxidative damage to tumors. The manuscript can be considered for publication after addressing the following major issues:

1. Since both Fe³⁺ and gallate are small in size, what is the advantage of choosing hollow silica over mesoporous silica?
2. The authors demonstrated the formation of GA-Fe structure in solution. How to directly prove the formation of the GA-Fe nanocomplex delivered by hollow silica nanocarrier intracellularly, considering that intracellular microenvironment is more complicated?
3. In addition, how to directly prove that the GA-Fe nanocomplex promotes two steps of sequential one-electron ORRs generating hydrogen peroxide intracellularly?
4. There are intrinsic Fe³⁺ presence in cells. Would such Fe³⁺ participate in GA-Fe nanocomplex formation intracellularly?
5. From Figure 7d, the tumors still show a growth trend after treating with FHPG for 14 days. The FHPG nanosystem with such complicated therapeutic pathway (delivery, dissociation, GA-Fe formation, and catalysis) seems not efficient in terms of cancer therapy when compared with other nanomedicine.
6. Key liver function index and blood chemistry metrics before and after treating with the nanosystem in vivo should be measured to further support the biosafety.

Reviewer #3:

Remarks to the Author:

In this manuscript, a nanomedicine for cancer therapy by loading gallate in a Fe-engineered hollow MSN functionalized with PEG was constructed. In situ, a nano-dimensional hexacoordinated GA-Fe complex was formed to promote the two-electron reduction of O₂ into H₂O₂ and further promote the generation of highly oxidizing •OH. The overall study is complete. This paper is recommended for publication after addressing following issues:

1. The water-soluble drug gallate is loaded in mesoporous silica without blocking the hole. Does the drug leak through the blood circulation? Will this leakage cause side effects?
2. What is the specific concentration of the drug and the Fe iron content in FHPG?
3. The drug release experiment in response to the acidity should be provided.
4. In Figure 7a, the blood circulation half-time of FHPG was calculated to be 1.75 h. Without appropriate reference value, what can "1.75h" represent?
5. In Figure S26, Supporting Information, the safety of FHPG on mouse HC11 cells was confirmed and the authors ascribe the low toxicity of nanoparticles in the HC11 cells to the neutral environment in normal cells was not conducive to Fe release. However, is there any possibility of iron ions could be endocytosed and then degraded in the acidic lysosome?
6. In Figure 4h, does the "Fe 2p_{1/2}" mean the zero-valent iron peaks?

Response to reviewer I.

Comments from reviewer I:

The authors describe the synthesis of a GA/Fe Fe-engineered mesoporous silica nanocarrier for the treatment of cancer. The authors provide comprehensive materials characterization using multiple experiential approaches.

They draw upon a well know chemical reaction that is attributed to the breakdown of written works. Their approach to use the underlying chemistry responsible for this to treat cancer tumours is very novel. The authors provide clear and scientifically sound discussions and conclusions based on their experimental data at all stage of materials development and cellular evaluation.

Please provide some clarification in the text regarding the possible dangers of the mesoporous material degrading at neutral pH. Whilst this degradation is shown to be slow, what is the likely hood that the nanocarrier could breakdown outside the tumour environment and cause damage to other organs. The research does not use a tumour targeted approach and relies on the vascular system to deliver the material to the tumour, therefore off target effects would be likely.

Many of the figures contain data with error bars. Are these standard errors, standard deviations or something else? Also there was no statistics provided on the significance of data variation.

The level of experimental detail was sufficient for a researcher to reproduce the work.

This paper provides novel method to cancer tumour treatment and I feel the work will be of interest to researchers and clinicians working in the oncology field. I also feel the comprehensive materials synthesis and characterization will be of much interest to a wide materials research community.

Response: Thank you very much for the positive comment and kind recommendation. According to your suggestion, the possible danger of nanomedicine at neutral pH has been provided in the revised manuscript (Page 31).

All the error bars in this work are standard deviations, and the related statements have also been

supplemented in the captions of corresponding figures. The necessary significances of data variations in this work have also been calculated and marked in the corresponding figures according to your suggestion (Figure 5g, Figure 6e and 6f, Figure 7d and 7e, Figure S28 and S29).

Response to reviewer II.

Comments from reviewer II:

In this study, the authors prepared a nanosystem for cancer therapy by loading gallate into a Fe-engineered hollow silica nanocarrier. Intracellularly released Fe^{3+} and gallate can trigger the coordination reactions with each other to form GA-Fe nanocomplex in situ, leading to oxygen reduction reaction for generating hydrogen peroxide. In addition, GA-Fe as a Fenton-like agent can catalyze hydrogen peroxide into hydroxyl radical to provide oxidative damage to tumors. The manuscript can be considered for publication after addressing the following major issues:

Response: Thank you very much for the comment and kind recommendation. Please find the following detailed responses to your suggestions.

1. Since both Fe^{3+} and gallate are small in size, what is the advantage of choosing hollow silica over mesoporous silica?

Response: Thank you very much for the kind question. The therapeutic approach in this work was achieved by doping Fe element in the framework of mesoporous silica while loading gallate in the cavity. However, till now, only Fe-doped hollow silica nanoparticles with monodispersity have been prepared in our research field while no mature technology has been developed to synthesize monodispersed and non-hollow mesoporous silica nanoparticles with significant Fe doping. The hollow structure not only endows the nanocarrier with high gallate-loading capability, but also promotes the degradation of the

silica nanoparticles.

2. The authors demonstrated the formation of GA-Fe structure in solution. How to directly prove the formation of the GA-Fe nanocomplex delivered by hollow silica nanocarrier intracellularly, considering that intracellular microenvironment is more complicated?

Response: Thank you very much for the kind question. In fact, we have also tried to provide more characterizations on the intracellularly formed GA-Fe nanocomplex in addition to the bio-TEM image of HeLa cells in Figure 6b. According to your question, we have further provided the high-resolution TEM image of a single GA-Fe nanoparticle in cells with a high crystallinity evidencing its intracellular formation (Figure S26 in the revised supporting information). The further compositional characterization of GA-Fe in cells is indeed difficult and cannot be achieved based on the currently available characterization technique.

3. In addition, how to directly prove that the GA-Fe nanocomplex promotes two steps of sequential one-electron ORRs generating hydrogen peroxide intracellularly?

Response: Thank you very much for the constructive question. According to your question, we have used the adenovirus encoding SOD or catalase to infect HeLa cells before FHPG treatment to deplete the intracellularly generated superoxide anion or H₂O₂, respectively (Page 39 in revised manuscript), and much less •OH was detected based on the result of flow cytometry (Figure S28 in revised supporting information), demonstrating the GA-Fe nanocomplex can promote two steps of sequential one-electron ORRs for generating superoxide anion and H₂O₂ intracellularly. The related discussions have also been supplemented in the revised manuscript (Page 26 and 27).

4. There are intrinsic Fe³⁺ presence in cells. Would such Fe³⁺ participate in GA-Fe nanocomplex formation intracellularly?

Response: Thank you very much for the kind question. We cannot totally exclude the possibility of the reaction between endogenous intracellular Fe^{3+} and the delivered gallate. However, as the concentration of endogenous Fe^{3+} is relatively low (3.5-230 μM according to the literature: Free Radical Biol. Med. 2013, 65, 143-149), the generation of GA-Fe nanocomplex mainly rely on the delivered Fe^{3+} engineered in the nanocarrier.

5. From Figure 7d, the tumors still show a growth trend after treating with FHPG for 14 days. The FHPG nanosystem with such complicated therapeutic pathway (delivery, dissociation, GA-Fe formation, and catalysis) seems not efficient in terms of cancer therapy when compared with other nanomedicine.

Response: Thank you very much for pointing out this issue. The tumor therapeutic efficiency relies not only on the intrinsic properties of nanomedicines, but also on other specific experiment factors such as injected dose and tumor subtype. Therefore, it may be not easy and appropriate to directly compare the therapeutic efficacies of different nanomedicines reported in different sources. We expect that the therapeutic efficacy of FHPG can be further improved substantially during its future anticancer applications, based on the current relatively high antitumor efficacy that significantly slows down tumor growth. Fortunately, thanks to the negligible toxicity of the nanomedicine compared to the conventional highly toxic chemodrugs, the dosages of the nanomedicine can be largely elevated, which is expected to show much better anti-tumor performance

In this study, distinct enhancement of therapeutic efficacy can be observed after the combination of nanocarrier and gallate in one single nanosystem, providing a feasible strategy to significantly promote oxidative damage of tumor, by the well-designed sequential reactions favoring the generation of large amount of $\bullet\text{OH}$. We hope that such a design of nanomedicine can inspire future works in this field.

6. Key liver function index and blood chemistry metrics before and after treating with the

nanosystem in vivo should be measured to further support the biosafety.

Response: Thank you very much for the kind suggestion. The key hepatic function index and blood chemistry metrics before and after treating with the nanosystem have been provided in the supporting information (control group and FHPG group in Figure S35 of revised supporting information), which demonstrate the satisfactory biocompatibility of the nanomedicine.

Response to reviewer III.

Comments from reviewer III:

In this manuscript, a nanomedicine for cancer therapy by loading gallate in a Fe-engineered hollow MSN functionalized with PEG was constructed. In situ, a nano-dimensional hexacoordinated GA-Fe complex was formed to promote the two-electron reduction of O₂ into H₂O₂ and further promote the generation of highly oxidizing •OH. The overall study is complete. This paper is recommended for publication after addressing following issues:

Response: Thank you very much for the positive comment and kind recommendation. Please find the following detailed responses to your suggestions.

1. The water-soluble drug gallate is loaded in mesoporous silica without blocking the hole. Does the drug leak through the blood circulation? Will this leakage cause side effects?

Response: Thank you very much for the kind questions. We cannot exclude the possibility of potential gallate leakage through the blood circulation. In fact, it also an unsolved issue encountered by all the currently-published non-blocked drug delivery systems. However, as the gallate is mainly loaded in the inner cavity of the hollow nanocarrier, while the shell of the nanocarrier is relatively thick, it can be inferred that the leakage of gallate is not significant.

Based on our experimental results such as hematological and histological evaluations (Figure S35-37 in the revised supporting information), this minor leakage will not trigger distinct side effect, which may be attributed to the intrinsic good biocompatibility of gallate.

2. What is the specific concentration of the drug and the Fe iron content in FHPG?

Response: Thank you very much for the constructive question. The mass percentages of Fe and gallate in FHPG are 5.707 wt% and 16.817 wt%, respectively, for keeping the stoichiometric ratio of Fe to gallate to be 1:1. We have further supplemented the data in the revised manuscript according to your question (Page 34).

3. The drug release experiment in response to the acidity should be provided.

Response: Thank you very much for the kind suggestion. In fact, we have also ever tried to directly investigate the release kinetics of gallate in acidic environment. However, as the released gallate will react immediately with the co-released Fe^{3+} to form GA-Fe nanocomplex, which is hard to be separated from the nanomedicine precipitate after centrifugation, therefore we are sorry to say, it is indeed hard to directly quantify the concentration of released gallate in the solution.

4. In Figure 7a, the blood circulation half-time of FHPG was calculated to be 1.75 h. Without appropriate reference value, what can “1.75h” represent?

Response: Thank you very much for pointing out this issue. The blood circulation half-time of FHPG is roughly equivalent to that of PEGylated pristine MSN (1.85 h) according to our previous report on the pharmacokinetics of spherical MSNs (small 2011, 7, 271-280). According to your question, we have revised the related discussion and cited the literature in the revised manuscript (Page 29, Ref 42).

5. In Figure S26, Supporting Information, the safety of FHPG on mouse HC11 cells was confirmed and the authors ascribe the low toxicity of nanoparticles in the HC11 cells to the neutral environment in normal cells was not conducive to Fe release. However, is there any possibility of iron ions could be endocytosed and then degraded in the acidic lysosome?

Response: Thank you very much for pointing out this issue. We cannot exclude the possibility of potential nanomedicine degradation in the acidic lysosome. However, the abundant hydrolase in lysosome can equally degrade the released gallate or the formed GA-Fe. Therefore, according to our experimental results, the low toxicity of nanoparticles in the HC11 cells means that acidic lysosome degradation induced toxicity is insignificant in this work.

6. In Figure 4h, does the “Fe 2p_{1/2}” mean the zero-valent iron peaks?

Response: Thank you very much for the kind question. In fact, it is also a merged peak between Fe (II) and Fe (III). However, usually researchers do not split this peak, as the investigation on the peak of Fe 2p_{3/2} is much more meaningful and typical based on the fundamental mechanism of XPS.

Reviewers' Comments:

Reviewer #1:

Remarks to the Author:

The authors have addressed my comments to my satisfaction. I feel that this manuscript is now suitable for publication.

Reviewer #2:

Remarks to the Author:

As the authors have well addressed reviewers' comments, the revised manuscript is recommended for publication.

Reviewer #3:

Remarks to the Author:

All the concerns have been addressed and it is acceptable now.